# Impacts of marine organic emissions on low level stratiform clouds – a large eddy simulator study

Marje Prank[1], Juha Tonttila[2], Jaakko Ahola[1], Harri Kokkola[2], Thomas Kühn[2], Sami Romakkaniemi[2], Tomi Raatikainen[1]

[1] Climate System Research Unit, Finnish Meteorological Institute, Helsinki, 00560, Finland
[2] Atmospheric Research Centre of Eastern Finland, Finnish Meteorological Institute, Kuopio, 70211, Finland

*Correspondence to*: Marje Prank (marje.prank@fmi.fi)

**Abstract.** The goal of this study is to investigate the role of organic aerosols emitted with sea spray or formed from marine gas phase emissions of volatile organic compounds (VOCs) in influencing the stability of stratiform marine clouds. We aim at pointing out the processes and drivers that could be relevant for global climate and should thus be considered in large scale models.

We employ UCLALES-SALSA, a large eddy simulator coupled with an aerosol-cloud microphysical model together with different parameterizations for emission of sea salt, primary organic aerosol and VOCs from sea surface and formation of secondary organic aerosol (SOA) to simulate the conditions of the DYCOMS-II observational campaign characterized by low level stratocumulus clouds transitioning from closed cells to drizzling open cell structure.

We find that the inclusion of sea spray emissions can both extend and shorten the transitioning timescale between closed and open cells based on the parameterization employed. Fine sea spray provides extra cloud condensation nuclei (CCN) and delays the onset of drizzle as the collision-coalescence process is slowed down due to smaller cloud droplet mean size. The coarse mode has an opposite effect due to giant CCN (GCCN) speeding up the drizzle formation through the enhanced collision-coalescence processes. The balance between two process depends on the model parameterization employed. Compared to differences between different sea spray parameterizations, the sensitivity of the clouds to the variations in organic fraction of sea spray and hygroscopicity of the emitted particles is relatively limited. However, our results show that it is important to account for the size dependence of the sea spray organic fraction as attributing organic emissions to coarse mode noticeably reduces the GCCN effect. In addition, including the secondary organic aerosol (SOA) formation from VOCs can potentially have a noticeable impact, but only when emitting the highest observed fluxes of monoterpenes. This impact is also highly sensitive on the size distribution of the background aerosol population. SOA production from isoprene is visible only if aqueous phase SOA production pathways are included, and even then, the effect is lower than from monoterpenes.

# 1    Introduction

The 5th Assessment Report of IPCC, (2013) recognized aerosols and clouds as the dominant sources of uncertainty in climate projections and aerosol-cloud interactions remain challenging also for models in the 6th Assessment Report (Forster et al., 2021). They point out that in both Coupled Model Intercomparison Projects CMIP3 and CMIP5 aerosol-cloud interactions constituted the dominant source of inter-model differences and majority of this spread is attributable to differences in parameterizations for shallow clouds prevalent over oceans (Dufresne and Bony, 2008; Vial et al., 2013).

Meehl et al. (2020) and Zelinka et al. (2020) show that the same still holds for the recent CMIP6 ensemble and call for more research into aerosol-cloud interactions.

In addition to available water vapour, longwave radiative cooling and updraft velocity that determines the cooling rate and thus the supersaturation, cloud droplet formation depends on the concentration of the available cloud condensation nuclei (CCN) (e.g. Rosenfeld et al., 2019). The two major controlling factors of CCN concentration are aerosol number

concentration and aerosol size, while particle composition also can have an effect through the hygroscopicity (Bougiatioti et al., 2020). Thus, it is important to explicitly model the concentration, size distribution and composition of the CCN. Over the oceans the major source of CCN is sea spray (consisting of sea salt and organic species), and an additional noticeable contribution comes from secondary organic aerosol (SOA) formed from oxidation products of marine emitted volatile organic compounds (VOCs) (McCoy et al., 2015). It has been found challenging for models to reproduce the secondary and

primary organic aerosols over remote ocean areas (Hodzic et al., 2020).

In the current modelling study, we investigate how marine emissions of sea spray and VOCs affect the low-level liquid clouds and how the emission related uncertainties transfer to the prediction of cloud properties. Large uncertainties still exist in the parameterizations of sea spray flux, size distribution and composition (Naik et al., 2021). These uncertainties are especially large during warm, biologically active periods, as both water temperature and organic content have been shown to

impact the sea spray emission. Both of these factors are going to be impacted by the changing climate, but it is hard to distinguish these two effects when developing sea spray emission schemes for large scale models. Grythe et al. (2014) compared a number of sea spray emission schemes and showed that the emitted mass fluxes can differ by more than an order of magnitude with strong disagreements in emitted size distributions and the wind speed and sea surface temperature (SST) dependence of the emission. Observations and model-measurement comparisons support the growth of sea salt mass flux

with increasing SST (Jaeglé et al., 2011; Liu et al., 2021; Sofiev et al., 2011) and point out the importance of including it in model parameterizations. However, the case is not clear for the particle number flux that mostly consists of small particles. Forestieri et al. (2018) show strong disagreements between laboratory experiments and large-scale model-measurement comparison based parameterizations – while the latter tend to extend the mass flux based monotonic temperature dependence also to the number flux, the laboratory experiments show a different dependence with increase of particle number flux for the

low temperatures. For example, Mårtensson et al. (2003) show different temperature dependence for fine and coarse sea spray, with the fine mode that dominates the number flux having opposite temperature dependence from the coarser

particles. However, as shown by Barthel et al. (2019) and Forestieri et al. (2018), large uncertainties exist in the magnitude of this dependence.

Cochran et al. (2017) reported large diversity of molecules making up the organic fraction of sea spray, which varied depending on the prevalent phytoplankton and bacteria species in the surface water. Large fraction of these organic molecules are surface active and experiments conducted by Nielsen and Bilde (2020) demonstrated a complex interdependence of the effects of temperature and organic surfactants to particulate emission from breaking bubbles. Considering these complexities, it is not surprising that disagreements exist related to the impact of the organic content of the sea water. For instance, Modini et al. (2013) show suppression of aerosol emission from bursting of a single bubble by organic surfactant, while Fuentes et al. (2010) and Long et al. (2014) report significant increase of particle number flux with increased dissolved organic carbon (DOC) content from diatomaceous exudates, and Bates et al. (2020) and Mayer et al. (2020) did not see any significant effect of plankton bloom on primary sea spray emission flux. According to Lv et al. (2020) and Fuentes et al. (2010), the effect depends on the type of surfactant or algal exudate, which might explain these disagreements.

The organic content can also affect the particle hygroscopicity and thus its ability to act as CCN (Forestieri et al., 2016; Fuentes et al., 2011). Fuentes et al. (2011) estimated the hygroscopicity parameters of pure algal exudate particles in range of 0.062-0.164, which is substantially lower than that of sea salt (i.e. 0.91-1.33 for NaCl according to Petters and Kreidenweis (2007)). Cochran et al. (2017) showed that the hygroscopicity of the sea spray aerosol depended on the type of the organic molecules dominant in the sea spray, which varied between different sized particles. Size dependence of the composition has also been demonstrated by Ault et al. (2013), Facchini et al. (2008) and Kaluarachchi et al. (2022). However, some other recent publications (Bates et al., 2020; Collins et al., 2016; Cravigan et al., 2020) have reported that the hygroscopicity of sea spray aerosol seems to stay largely invariable regardless the organic fraction, while (Ovadnevaite et al., 2011) reported a complex behaviour of low hygroscopic growth in aerosol phase combined with high CCN activation efficiency.

Marine emission of VOCs plays an important role in controlling the size distributions of marine aerosol by condensing on ultrafine particles and growing them to CCN size (Burkart et al., 2017; Croft et al., 2019, 2020; Yu and Li, 2021). In fact, the role of secondary marine aerosol has even been found to dominate over that of primary (Mayer et al., 2020). Dimethyl sulfide (DMS) is considered the most important source of CCN in the clean ocean areas, others include alkanes, alkenes, aromatics, terpenoids and amines, halogenated organics and oxygenated volatile organic compounds (OVOCs). Large amount of research already exists on DMS while, with the exception of the terpenoids, not enough is currently known about the emission rates and chemistry of the other above-mentioned VOCs to include them in model simulations. Thus, in the current study we focus on two terpenoid species - isoprene and monoterpenes.

Isoprene has been detected in sea water and marine atmosphere (Kim et al., 2017; Rodríguez-Ros et al., 2020; Shaw et al., 2010; Yassaa et al., 2008). Its oxidation products have been observed in marine aerosol (Cui et al., 2019; Hu et al., 2013) and proposed to explain the observed variability in CCN concentration over Southern Ocean (Meskhidze and Nenes, 2006). Little observational data exists of monoterpenes in ocean water (Button and Jüttner, 1989; Hackenberg et al., 2017), although

their marine production has been confirmed (Yassaa et al., 2008) and relatively high air concentrations have been observed in marine atmosphere (Kim et al., 2017; Shaw et al., 2010; Yu and Li, 2021). Monoterpene oxidation products have been detected in marine aerosol (Cui et al., 2019; Hu et al., 2013), and while the oceanic emission of monoterpenes is generally estimated to be smaller than that of isoprene (Meskhidze et al., 2015; Yassaa et al., 2008), it can have larger impact due to larger SOA yields (Yu and Li, 2021).

However, the uncertainties in the terpenoid ocean to atmosphere fluxes are large and estimating their emissions either bottom-up from algal production or top-down from air concentrations leads to large discrepancy (Luo and Yu, 2010). Kim et al. (2017) measured sea to air fluxes of both isoprene and monoterpenes. On average the fluxes were low while substantially higher peak values were observed over biologically active areas with recently elevated nutrient content due to increased vertical mixing by wind or rising currents. This large variability can at least partly explain the challenges in estimating the emissions.

Clouds are influenced by covariations in air flows and aerosols in a noticeably smaller spatial scale than the current resolution of climate models. Large eddy simulators (LES) can resolve these scales and have proved useful for understanding climate relevant cloud feedbacks (e.g. Bretherton, 2015). In the current study we use the UCLALES-SALSA large eddy simulator with representation of aerosol-cloud-precipitation interactions (Tonttila et al., 2017), where we have included sea spray and VOC emissions from sea surface. Our goal is to investigate the role of primary and secondary marine aerosols in the stability of the liquid phase clouds. As a case study we apply the model to simulate the conditions of the DYCOMS-II observational campaign characterized by low level stratocumulus clouds transitioning from closed cells to drizzling open cell structure. In order to quantify the sensitivity of the transitioning timescale to the above discussed processes and parameters we perform multiple model simulations with different published parameterizations of sea spray and terpenoid emissions. We aim to point out the processes and drivers that can be relevant to climate in large scale and should thus be considered in climate models.

## 2    Methods

### 2.1 Model description

We use the UCLALES-SALSA model (Tonttila et al., 2017) that combines the large eddy simulator UCLALES (Stevens et al., 1999, 2005; Stevens and Seifert, 2008) with a detailed sectional model of aerosol microphysics SALSA (Kokkola et al., 2008, 2018) extended with cloud and precipitation processes. SALSA simulates microphysical processes such as coagulation, sedimentation, partitioning of water between vapour and liquid phases, activation of aerosol particles to cloud droplets and growth of droplets to form precipitation.

Aerosol is described by two externally mixed modes, each consisting of a set of size bins with unique chemical composition. Tracking the two externally mixed modes allows us to separately follow the fate of the background aerosol and sea spray emitted during the simulation. The aerosol particle bins are based on the dry diameter of the particles. In the

current study we used 15 bins ranging from 3 nm to 10 micrometres (sizes here and further on are given as dry diameter unless stated differently). This description is extended to also cover the cloud droplets so that particles that activate as cloud droplets are moved to a parallel bin structure identical to the one for aerosols. Condensation/evaporation and cloud droplet activation are computed according to Köhler theory (Köhler, 1936), taking into account the size and composition of particles in every size bin. Drizzle formation is parameterized using the autoconversion scheme developed by Seifert and Beheng (2001). In UCLALES-SALSA, the autoconversion schemes essentially move the cloud droplets that grow to drizzle-size from the dry-size based cloud bins to the wet-size based rain bins, allowing accurate modelling of precipitation formation via condensation and coalescence processes (Ahola et al., 2021). The parameterization is a computationally efficient alternative to explicitly simulating the initial rain formation step based on collisions between cloud droplets (Tonttila et al., 2021). We use 7 rain bins covering wet diameters from 50 μm to 2 mm. For every bin (aerosol, cloud and rain) the number concentration, the amount of all chemical species and water amount are tracked. All microphysical processes (condensation, evaporation, coagulation/coalescence, sedimentation etc) are simulated identically for aerosol, rain and cloud droplets. The model structure and applied algorithms are explained in more detail by Tonttila et al. (2017).

Several different sea spray emission parameterizations were included in the model. In various simulations in this study (Table 1) we make use of the emissions parameterizations by Mårtensson et al (2003), Gong (2003), Fuentes et al (2010) and Monahan et al (1986). In majority of the presented simulations (case names starting with F10 in Table 1) the emission fluxes for submicron (dry) particles are modelled following Fuentes et al (2010) who parameterized the impact of sea water DOC content on emitted aerosol number size distribution. As Fuentes et al (2010) do not provide scaling from their experimental setup to emission from open ocean surface, we have scaled their flux to reproduce the number flux of Mårtensson et al (2003) in the size range covered by both parameterizations (20-450 nm) at our base case sea surface temperature of 10°C by multiplying it with 5500. The temperature dependence of submicron sea spray emission for this parameterization is implemented following Forestieri et al. (2018), the smaller modes having no temperature dependence and largest one increasing linearly. The flux of supermicron particles is computed following Monahan et al (1986). To study the impact of the uncertainties in the sea spray emission parameterizations, extra simulations are made using the temperature dependent scheme of Mårtensson et al (2003) extended with the Monahan et al (1986) scheme for sizes coarser than 2.8 micron (M03 in Table 1) and Gong (2003) scheme (G03 in Table 1). Monahan et al (1986) and Gong (2003) schemes are coupled with temperature dependence from Jaeglé et al (2011). For sea salt the dry diameter is assumed to be half of that at 80% relative humidity, as the error of this relation was reported to be below 5% by Sofiev et al. (2011).

The selected sea spray flux size distributions are shown in Figure 1. As the Fuentes et al (2010) scheme (black) was scaled to reproduce the flux of Mårtensson et al (2003) at 10° C (MO3, green line), the emissions by these schemes are relatively similar. The parameterization by Gong (2003) (G03, blue line) was included in this study to cover the wide range of fluxes predicted by various sea spray emission parameterizations, as it differs from the other used parameterizations by much lower emission of small particles. As seen from comparing the black (F10) and brown (F10-DOC=512μM) lines on Figure 1, the impact of the largest concentration of dissolved organic carbon considered by Fuentes et al (2010) (512μM) on

the emission flux of the Aitken mode sea spray is comparable in size with the temperature effect shown for the Mårtensson et al (2003) parameterization (light to dark green), while there is almost no effect on the rest of the size range.

The organic fraction of the emitted particles is computed following the windspeed, particle size and Chlorophyll alpha (Chl α) dependent parameterization of Gantt et al (2011). Experiments with nonzero Chl α start with G11 in Table 1. As the relation between Chl α and DOC in sea surface is complex (Fuentes et al., 2010; Van Pinxteren et al., 2017), this scheme is applied independently of the DOC dependent sea spray flux parameterization. When using it, no changes are made to the total sea spray flux or the emission size distribution, but for each size bin part of the sea spray is emitted as primary organic matter instead of sea salt. Internal mixing of sea salt and organic matter is assumed. Hygroscopicity parameter of pure sea salt is set to 1.0 (Christiansen et al., 2020) and 0.1 is used for organic matter, which is in the range reported by Kuang et al. (2020) for SOA and by Fuentes et al. (2011) for algal exudate and only slightly lower than what was reported by Mayer et al. (2020) for marine organic species. The effect of organic content on particle hygroscopicity is demonstrated by the yellow lines on Figure 1. With high chlorophyll alpha concentration (2 µg/L) the small particles up to 0.3 microns in diameter consist almost entirely of organic matter while the large ones stay almost free of organics, leading to large difference in their hygroscopicities. Primary marine organics are treated as nonvolatile.

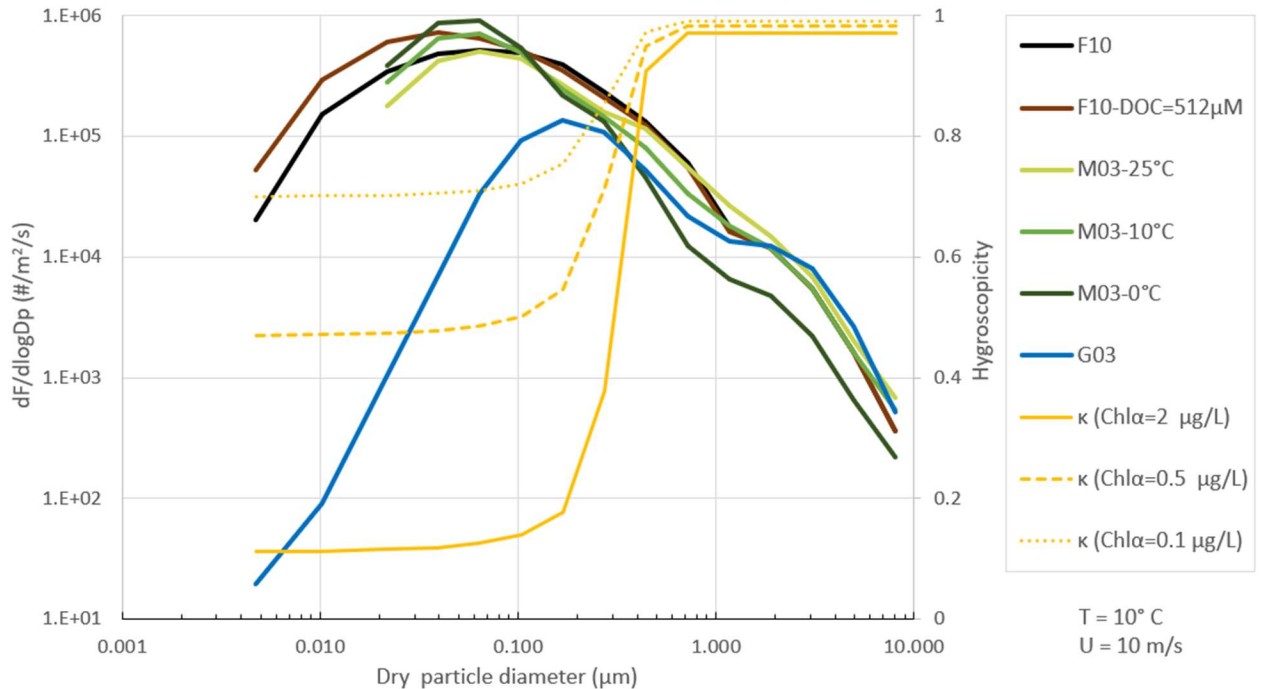

Figure 1. Sea spray flux from different parameterizations and hygroscopicity parameter kappa derived based on the size dependent organic fraction for different Chlorophyll alpha concentrations in sea water. Fluxes are shown for pure sea salt. Kappa values of 1 and 0.1 are assumed for the sea salt and the organics respectively.

Partitioning of semivolatile organic species between aerosol and gas phase has been implemented using the Volatility Basis Set approach (VBS, Donahue et al 2006, Robinson et al 2007)). VBS is coupled to SALSA identically to what was used by Mielonen et al. (2018). We use three volatility bins with saturation concentrations 0, 1, and 10 µg/m$^3$. Our approach deviates slightly from the classical implementations of Volatility Basis Set (VBS) (e.g. Farina et al. (2010)) that assume instant equilibrium between the gas and aerosol phases. In particular, we first use the VBS framework to compute the equilibrium vapour pressure of each semivolatile species for every aerosol, cloud and precipitation size bin according to their organic content. These values are then used for computing diffusion-limited condensation and evaporation using the analytical predictor of condensation scheme (Jacobson, 1997).

As the study concentrates on clean marine areas, we focus on biogenic isoprene and monoterpenes as SOA precursors. Their fluxes from marine surface can either be prescribed or computed from their concentrations in sea water following Wanninkhof (2014).

Isoprene and monoterpenes are oxidized by the OH and NO$_3$ radicals and ozone using reaction rates from Atkinson et al. (2006). The concentrations of the gas phase oxidants are prescribed at levels representative of longer term averages and kept constant over the simulation period of 12 hours and their emission, production and consumption are not explicitly modelled. The stoichiometric coefficients give the yields of the different semivolatile organic species (described by the volatility bins) resulting from precursor oxidation. The coefficients for monoterpenes originate from Kokkola et al. (2014) and for isoprene these are derived from the values used by Henze and Seinfeld (2006). The oxidation reactions together with the stoichiometric coefficients are listed in Table S1. Further oxidation of the semivolatile VBS species towards lower volatility bins is left out of these relatively short simulations. Due to low water solubility of the fresh SOA, the impact of liquid water on partitioning is ignored.

The model also includes an additional aqueous phase SOA formation route. We use a simplified model for the aqueous formation of SOA from IEPOX and glyoxal, similar to what was used by Mielonen et al. (2018). For partitioning the IEPOX and glyoxal to aqueous phase we account for the liquid water amount, and to take into account the "salting-in" effect (Kampf et al., 2013) we use higher effective Henry's law constants for aerosol than for cloud and rain droplets. For glyoxal we apply the effective Henry's law constants from Kampf et al (2013). The ionic strength has been reported to have an impact also on IEPOX uptake (Gaston et al., 2014) but its magnitude has a large uncertainty. Thus, for aerosol phase we use the measurements of Nguyen et al. (2014), and for cloud droplets and rain we have lowered the value to $10^5$ M atm$^{-1}$, same as used by e.g. Jo et al. (2019). As the background aerosol defined by Ackerman et al. (2009) for this case study is a simplified proxy that consists of pure ammonium bisulfate and does not reflect the complexity of real marine aerosol, we do not account for particle composition in more detail. The reactive partitioning model does not include any aqueous phase chemical reactions or irreversible processes. Gas phase oxidation reactions of IEPOX and glyoxal are included and are listed in Table S1 in the Supplemental Material. To reduce the computational demands the aqueous processes were turned on only in a dedicated sensitivity simulation.

## 2.2 Model experiments

The parameters for the LES simulations are based on the second research flight of the DYCOMS II nocturnal aircraft campaign near Californian coast in July 2001 (Stevens et al., 2003), characterized by lightly drizzling deck of closed-cell stratocumulus transitioning to heavily drizzling open cells. Additional CCN can substantially weaken the precipitation and retain cloud water in weakly precipitating marine boundary layer making it more sensitive to changes in aerosol concentration than strongly precipitating or non-precipitating regimes (Wang et al., 2011). Ackerman et al (2009) adapted the DYCOMS II case for LES intercomparison and demonstrated its aerosol sensitivity. The case has been previously used by Tonttila et al. (2017) for evaluating the UCLALES-SALSA model, demonstrating good skill to capture the cloud properties. Apart from including the marine emissions, the model setup and inputs used for the current study are almost identical to those used by Tonttila et al. (2017), with only a couple of small differences. Firstly, the model domain has been enlarged from 6 to 10 km to better represent the scale of the open cell structures. While by the end of the simulations the size of the open cells exceeds 10 km (Figure S1), preliminary tests showed that the 10 km domain allowed to simulate the transition process in this case with sufficient accuracy (Figure S2). Secondly, to reduce computational demands, the radiative transfer parameterization of Ackerman et al., (2009) is used instead of the 4-stream scheme. Thirdly, the original autoconversion parametrization used by Tonttila et al. (2017) is replaced by the double-moment scheme by Seifert and Beheng (2001). Although the onset of precipitation is seen earlier with the time-independent autoconversion approach than with the collision-based rain formation, the schemes produce consistent precipitation rates. Sensitivity of the results to the autoconversion scheme is shown in the Supplementary Material, Section 1.3.

The model domain covers 10 x 10 km area with 60 m horizontal resolution. Vertically the domain consists of 97 layers reaching up to 1.5 km, layer thickness increasing from 5 m at the surface to 80 m at the model top. The resolution, optimized for this case by Ackerman et al. (2009), is less than 25 meters for all in-cloud and below-cloud layers and 5 meters in the regions with largest gradients (near surface and cloud top). The 12 hours long simulations were run with 1 s timestep. If needed, the timestep is automatically reduced during the simulation to maintain the numerical stability of the transport processes. Within a single model timestep, the condensation equation and thus water partitioning between gas and liquid phases is solved in sub-timesteps of 0.05s to avoid problems with small aerosol particles that quickly respond to changes in conditions.

For each grid cell, the background aerosol was initialized with a bimodal distribution of ammonium bisulfate aerosol defined by Ackerman et al (2009). The first hour of the simulations is used as a spin-up period to allow the turbulence to build up and boundary layer to settle. In addition to drizzle formation, also the marine emissions are inactive during spin-up to avoid build-up of emitted components in the lowest model layers while the turbulence is still developing. Horizontally averaged vertical profiles and domain average parameters are saved every 60 seconds; 3D fields are saved once per hour.

Affecting the cloud processes by changing CCN number and properties could lead to feedbacks to the sea spray emission through changing wind fields. This kind of feedback processes are out of the scope of the current study but could mask or

amplify the direct effects of the studied parameters. Thus, to reduce the impact of model feedbacks the sea spray emission
module is forced by constant windspeed of 10 m/s, selected to match the midpoint of the maximum annual sea spray
production range (7-16 m/s) reported by Grythe et al (2014). Table 1 and 2 list the 26 model simulations discussed in this
paper, for the sea spray and VOC emissions respectively. The simulations are set up to evaluate one effect at time in order to
be able to directly compare the magnitude of the different effects. The impact of specific changes to the emission fluxes is
evaluated by comparing with the no-emission control and the F10 experiment with SST 10 °C and no DOC, that we refer to
below as the base case. The no-emission control was run without any sea spray or VOC emission (NoEms in Table 1 and 2).
This simulation is almost identical to the one described by Tonttila et al (2017), with the above mentioned differences in
selected parameterizations.

The results regarding the primary sea spray flux parameterization and its temperature dependence are discussed in section
3.2, while section 3.3 concentrates on the impacts of sea water organic content either by flux enhancement or sea spray
composition. Unless specified otherwise in Table 1, the setup and parameters identical to the base case were used for all
260 other sea spray simulations discussed in these sections.

**Table 1. Model experiments for sea spray emission**

| Experiment | Sea spray emission | Sea spray SST dependence, SST (°C) | DOC (µM) Chl α (µg/L) | Section |
|---|---|---|---|---|
| NoEms | 0 | - | - | |
| **F10** | *F10, M86* | *F18, J11* 0, **10**, 25 °C | - | 3.2 Sea spray flux parameterization and its temperature dependence |
| G03 | *G03* | *J11*, 10 °C | - | |
| M03 | *M03, M86* | *M03, J11* 0, 10, 25 °C | - | |
| F10-DOC=512µM | *F10, M86* | *F18, J11, 10 °C* | DOC 512 Chl α 0 | 3.3 Impacts of marine organic carbon and assumptions about hygroscopicity |
| F10-G11-Chlα2=µg/L | *F10, M86, G11* | *F18, J11,* 10 °C | DOC 0 Chl α 2 | |
| F10-G11-Chlα=2µg/L -noSizeDep | *F10, M86, G11* | *F18, J11,* 10 °C | DOC 0 Chl α 2 | |
| *A09* – (Ackerman et al., 2009), *F10* – (Fuentes et al., 2010), *M86* – (Monahan et al., 1986), *F18* – (Forestieri et al., 2018), *J11* – (Jaeglé et al., 2011), *G03* – (Gong, 2003), *M03* – (Mårtensson et al., 2003), *G11* – (Gantt et al., 2011). **Base case shown in bold**. | | | | |

The model simulations with isoprene and monoterpene emissions are listed in Table 2. In order to evaluate the impact of
265 these VOCs we utilized the measurements of Kim et al (2017), who during the HiWinGS measurement campaign measured
the air concentrations and marine fluxes of both isoprene and monoterpenes over Northern Atlantic.

In majority of cases the VOC concentrations in the model were initialized with steady state mass mixing ratios reached by the model at cloud level (1 ng/kg of monoterpenes and, 2 ng/kg of isoprene) by the end of a 5-day long preliminary simulation on reduced horizontal grid. VOC fluxes in this preliminary simulation where kept constant at the mean levels

observed by Kim et al (2017) (0.83 pmol/m²/s of isoprene and 0.44 pmol/m²/s of monoterpenes) and oxidant concentrations were prescribed at levels representative of clean marine areas (20 ppb of ozone, 0.1 ppt of OH and 0.01 ppt of NO3 radicals, marked as "default" in Table 2). Boundary layer ozone concentrations close to this were measured during DYCOMS II campaign (UCAR/NCAR, 2006). Similar ozone and OH concentrations were also used by Kim et al (2017) in their calculations.

Section 3.4 discusses the changes in the cloud layer due to the SOA formed from the measured mean and maximum VOC fluxes and concentrations. The sensitivity of the SOA cloud effects to the background aerosol is discussed in Section 3.5.

**Table 2. Model experiments for VOC emission**

| Experiment | Aerosol initial condition | VOC initial condition (ng/kg) | VOC emission (pmol/m²/s) | Oxidants (ppb) | Section |
|---|---|---|---|---|---|
| NoEms | A09 | 0 | 0 | - | |
| K17-meanFlx | A09 | Isop: 2 Mtrp: 1 | Isop: 0.83 Mtrp: 0.44 | Default: O3 20 OH $10^{-4}$ NO3 $10^{-5}$, | 3.4 SOA formation from isoprene and monoterpenes |
| K17-meanFlx-meanInic | A09 | Isop: 23.5 Mtrp: 85 | Isop: 0.83 Mtrp: 0.44 | | |
| K17-meanFlx-medInic | A09 | Isop: 23.5 Mtrp: 15 | Isop: 0.83 Mtrp: 0.44 | hiOx: O3 40 OH $10^{-3}$ NO3 $10^{-4}$ | |
| K17-maxFlx | A09 | Isop: 2 Mtrp: 1 | Isop: 18 Mtrp: 27 | | |
| K17-onlyIsop | A09 | Isop: 2 Mtrp: 0 | Isop: 18 Mtrp: 0 | hiOx | |
| K17-onlyMtrp | A09 | Isop: 0 Mtrp: 1 | Isop: 0 Mtrp: 27 | hiOx | |
| K17-onlyIsop-aqSOA | A09 | Isop: 2 Mtrp: 0 | Isop: 18 Mtrp: 0 | hiOx | |
| K17-maxFlx-aerInic=crs | A09, no fine mode | Isop: 2 Mtrp: 1 | Isop: 18 Mtrp: 27 | hiOx | 3.5 Sensitivity to background aerosol size distribution |
| NoEms-aerInic=crs | A09, no fine mode | 0 | 0 | - | |
| K17-maxFlx-aerInic=sslt | F10 size distribution | Isop: 2 Mtrp: 1 | Isop: 18 Mtrp: 27 | hiOx | |
| NoEms-aerInic=sslt | F10 size distribution | 0 | 0 | - | |
| A09 - (Ackerman et al., 2009), F10 - (Fuentes et al., 2010), K17 - (Kim et al., 2017). | | | | | |

## 3    Results and discussion

### 3.1 Development of the cloud scene

Figure 2 illustrates the development of the cloud layer in the UCLALES-SALSA simulations. At the beginning of the simulation (left column) the clouds consist of closed cells with 1-2 km diameter and very little drizzle is observed. As the simulations progress (middle column), the size of these structures grows and light drizzle is formed. In the simulations without sea spray emissions (upper row) the clouds start to break after about 6 hours while the amount of drizzle increases. By the end of the run (right panel) the closed cells have turned into large open cells with heavy drizzle. The horizontal size of the structures grows from couple of kilometres to more than ten, exceeding the size of the 10 km model domain (Yamaguchi and Feingold, 2015). In the F10 case simulations that include sea spray (lower row) the changes are slower and by the end of the run the clouds have not yet fully reorganized.

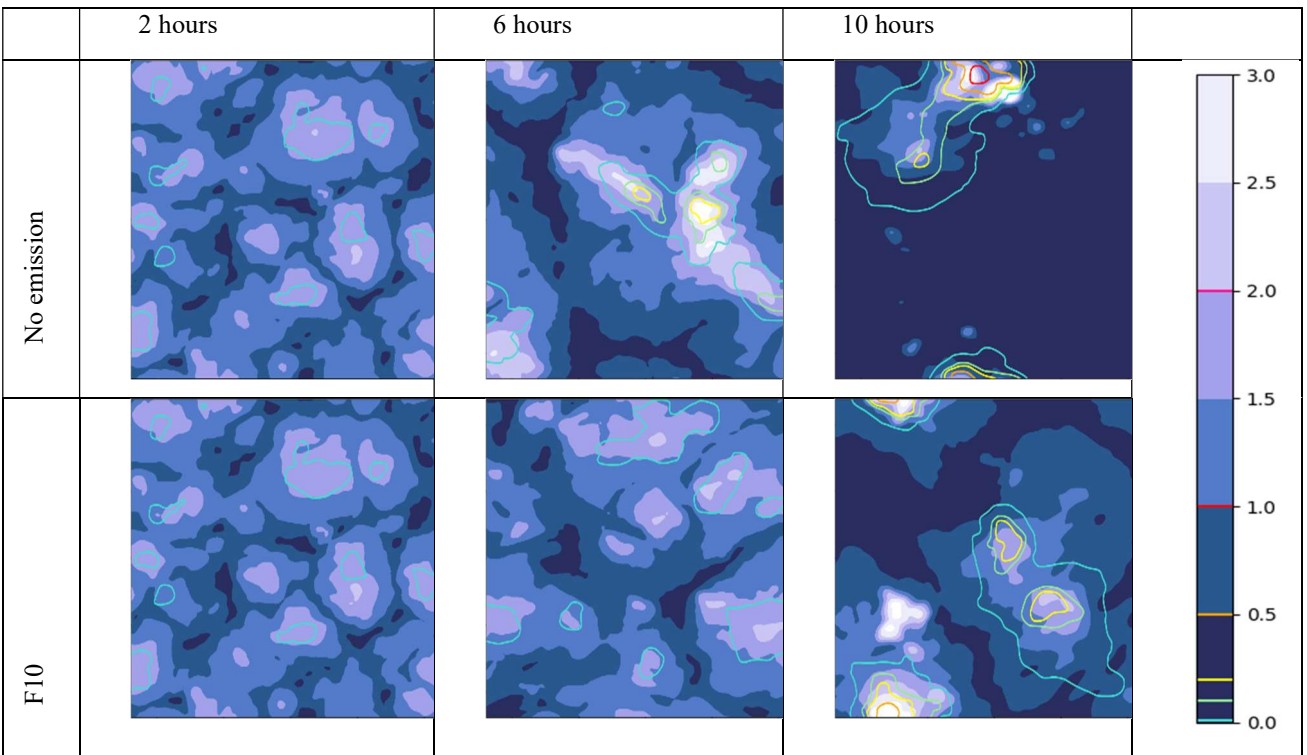

**Figure 2. Development of the cloud layer and drizzle in the 10x10 km model domain 2, 6 and 10 hours after the simulation start in the no-sea-spray (upper row) and F10 case (lower row) simulations. Contours – surface precipitation rate (mm/h), white shading –liquid water path (g/m$^2$) *0.01 (scaled to fit on same scale as precipitation).**

### 3.2 Sea spray flux parameterization and its temperature dependence

Comparing the no-sea-spray control run (grey) on Figure 3 with the base case (F10, black) shows that including the sea spray emission to the simulations can substantially increase the number of cloud droplets (dashed lines on panel a) and thus

lead to smaller cloud droplets (panel b) delaying the drizzle (panel e) and slowing down the removal of the background aerosol (panel f). This introduces a positive feedback to the system as the slower washout of the aerosol results in higher CCN concentration. As seen from the cloud water path plotted on panel c, this can delay the transition to open cells by several hours.

In order to evaluate the impact of uncertainties in sea spray emission, simulations were made with two alternative sea spray parameterizations: i) G03 - Gong (2003) and ii) M03 - Mårtensson et al (2003) extended with Monahan et al (1986) scheme for coarse sizes. While the emission flux for dry diameters below ~0.5 μm in the base case was scaled to match that of the M03 scheme (green) and supermicron flux in both cases is computed according to the same Monahan et al (1986) scheme, the different shapes of the size distributions (see Figure 1) still lead to somewhat different outcomes. The M03

scheme produces more particles with diameters below 0.1 μm and fewer larger ones, which leads to activation of large number of smaller particles resulting in smaller average cloud droplet size.

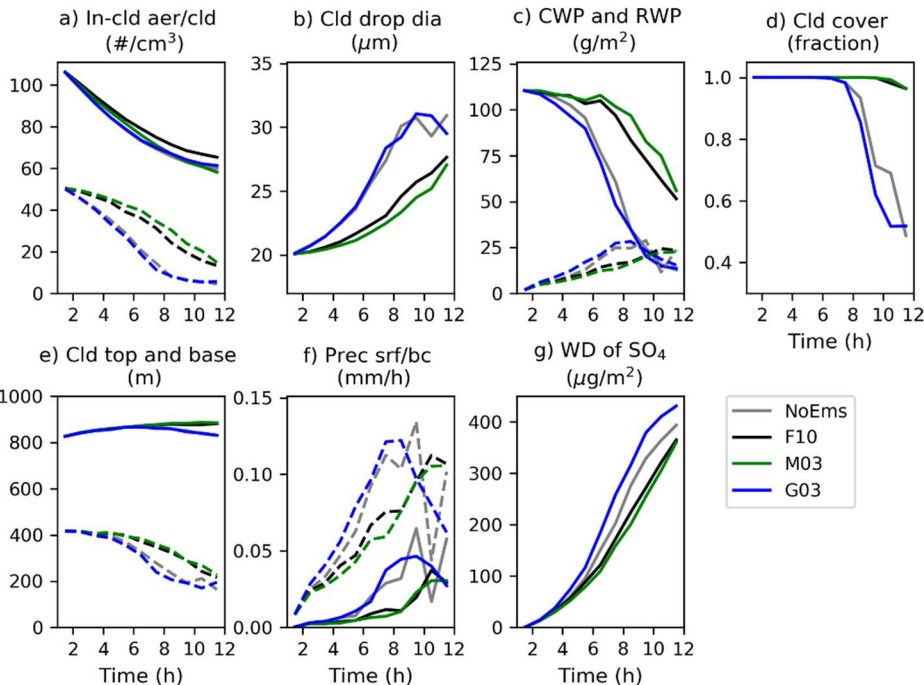

**Figure 3. Impact of sea spray flux on the cloud layer for different emission parameterizations. Hourly averaged time series,**
**mean over the model area. Panels: a – in-cloud interstitial aerosol (solid) and cloud droplet (dashed) concentration, b –cloud droplet size, mean over cloudy grid cells, c –Cloud liquid water path (solid) and rain water path (dashed), d -cloud fraction, e – height of cloud top (solid) and base (dashed), f –precipitation rate at surface (solid) and below cloud (dashed), g – cumulative wet deposition of background aerosol (ammonium bisulfate). Simulations: grey – no-emission control, black –F10, Blue – G03, green – M03 setups. All schemes were run with SST 10°C and 10 m/s windspeed. Grid cell is considered cloudy if cloud water mixing ratio**
**exceeds 1.e-5 kg/kg. All columns with at least 1 cloudy cell contribute to cloud fraction**

As the parameterization of G03 (blue) was originally developed to reduce the overestimation that was evident when extending the Monahan et al (1986) formula towards smaller sizes (Gong, 2003), its fine particle flux is significantly lower than the other parameterizations give for the same SST (about 5 times lower for 0.1 μm particles and more than an order of magnitude for smaller) while for the supermicron range it gives slightly higher flux (Figure 1). Compared with fine particles that extend the cloud lifetime, the effect of the coarse fraction is different - acting as giant CCN (GCCN) they speed up the collision-coalescence process that can lead to earlier drizzle formation (Houghton, 1938). For this reason, the cloud water path (panel c on Figure 3) reduces even faster with G03 emission (blue) than without any sea spray (grey).

The importance of temperature effects on sea spray emissions was tested by varying the SST only in the sea spray emission without affecting any other fluxes in the LES. Both the temperature dependency parameterization of Forestieri et al (2018) coupled with the Fuentes et al (2010) sea spray flux and Mårtensson et al (2003) temperature dependent parameterizations (F10 and M03 on Figure 4) were used. In addition to the 10°C base case, two alternative sea surface temperatures were simulated, covering the range from 0°C to 25°C.

As seen from the dark red and blue lines on Figure 4, for the case of the temperature dependency parameterization of Forestieri et al (2018) used in the base case setup with the Fuentes et al (2010) emission scheme that keeps the emission of fine modes constant regardless of SST, the effects are not large. The substantial differences between the light blue, green and red lines on Figure 4 demonstrate that in the Mårtensson et al (2003) scheme the temperature effects are much larger, as smaller number of fine particles and larger amount of coarse fraction is emitted with warmer SST (Figure 1), both working in the same direction of speeding up the transition. In the cold conditions of 0°C SST (light blue on Figure 4) the clouds only start to reorganize at the very end of the simulation.

As demonstrated by Terai et al. (2012), the ability of aerosols to suppress precipitation is relatively weak for clouds thicker than ~200 m. In our case the clouds start out about 400 m thick and indeed the drizzle increases and the cloud liquid water path starts reducing by the end of the simulation even with the case with the highest sea spray emission (M03 at 0 °C SST, light blue on Figure 4). In the base case about 1.5 million particles are emitted from a square meter of sea surface per second, around 2/3 of those in the size range to activate as CCN. The loss rate of particle number due to coalescence processes in the cloud is about 2.5 times higher than the emission of potential CCN, leading to continuous decrease in cloud droplet number. In the highest emission case (M03 at 0°C) the emission of potential CCN is only ~10% higher than in the base case, indicating that the noticeably longer cloud lifetime than in the base case could be partly also due to lower number of GCCN emitted in colder conditions.

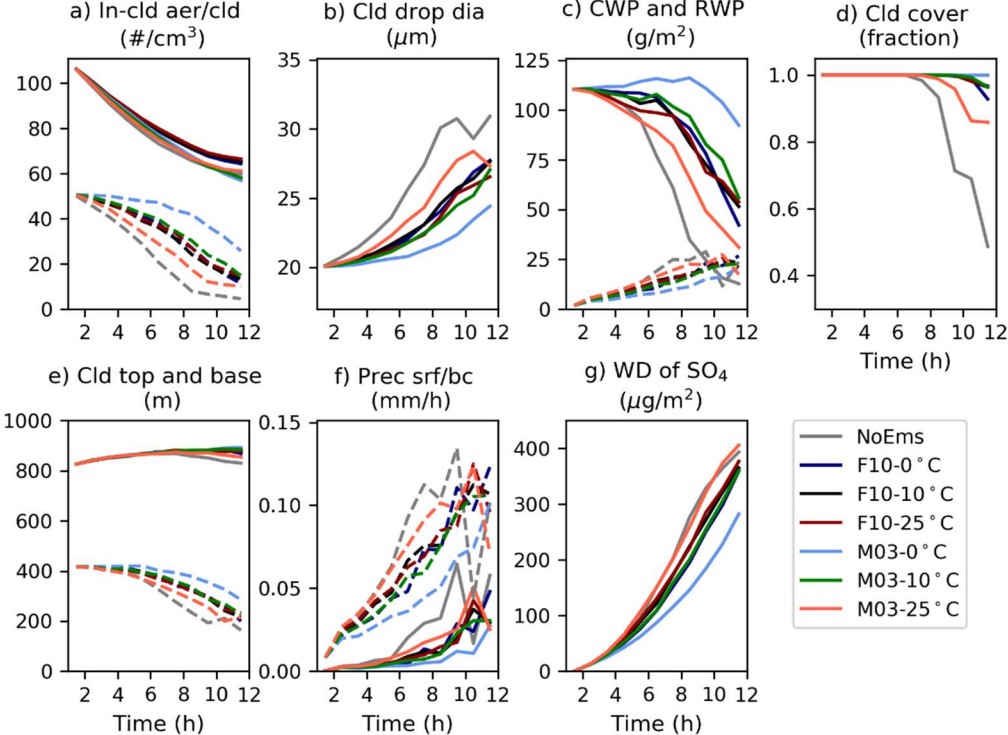

**Figure 4. Impact of sea surface temperature based on Forestieri et al (2018) (dark colours) and Mårtensson et al (2003) (light colours). Panels are the same as Figure 3. Temperatures from cold to warm: blue: 0°C, green/black: 10°C, red: 25°C**

### 3.3 Impacts of marine organic carbon and assumptions about hygroscopicity

Organic content of surface sea water can impact the aerosol by either changing the emission size distribution or changing the composition and thus the hygroscopicity of the emitted particles. To be able to directly compare the impacts these different pathways have on the cloud layer, we applied them separately in two different simulations. The impact of changing the size distribution was tested via a simulation (F10-DOC=512µM) otherwise identical to the base case but the dissolved organic carbon content in sea water was changed from zero to 512µM, the maximum value for the parameterization. This results in increased emission of sub 0.1 micron particles (Figure 1). In this simulation sea spray composition and resulting hygroscopicity was not changed. In order to evaluate the impact of changing hygroscopicity, another simulation (F10-G11-Chlα2=µg/L) was run with sea water Chlorophyll alpha concentration set to 2 µg /L and the organic fraction of the particles was computed by the Gantt et al (2011) aerosol size dependent scheme. In this simulation DOC was set to zero. The results of these simulations are shown on Figure 5 (flux enhancement from dissolved organic carbon in sea water with brown line and hygroscopicity reduction due to high organic fraction of small particles with yellow line). Both effects are relatively small compared to the temperature impacts of the M03 parameterization (Figure 4) and in the expected direction. The larger fine mode emission flux due to DOC leads to more cloud drops activating and a slight delay in sub-cloud precipitation (dashed lines on panel e) and thus slightly longer lifetime for the closed cell structures. The reduction of fine particle

hygroscopicity (yellow line) has somewhat more noticeable effect and expectedly in the opposite direction - reduced hygroscopicity of the fine aerosol leads to smaller number of particles activating, earlier drizzle and loss of clouds.

In order to evaluate the importance of the particle size dependence of the organic fraction, an extra sensitivity simulation
(F10-G11-Chlα=2μg/L -noSizeDep) was performed where the maximum organic fraction was attributed to emitted sea spray of all sizes (orange line on Figure 5). Lower water uptake of the coarse sea spray due to its large organic fraction substantially reduces the GCCN effect keeping the cloud field stable noticeably longer.

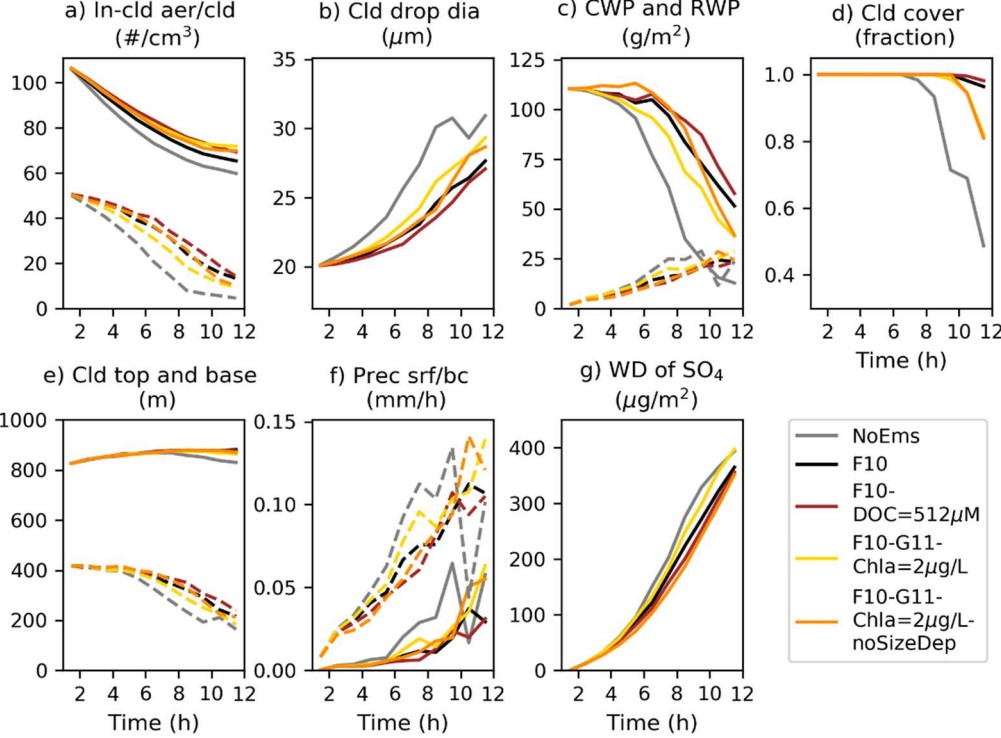

**Figure 5. Impact of dissolved organic carbon dependent enhancement of emission flux (brown) and organic aerosol fraction**
**impact through hygroscopicity change (yellow – size dependent scheme, orange – maximum organic fraction applied to all sizes).**
**Panels are the same as Figure 3.**

### 3.4 SOA formation from isoprene and monoterpenes

In this section we investigate the impact of secondary organic aerosol formed from marine isoprene and monoterpenes emissions on the cloud layer. The simulations are listed in Table 2.

The mean fluxes observed by Kim et al. (2017) during the HiWinGS measurement campaign over the Northern Atlantic (0.83 pmol/m$^2$/s of isoprene and 0.44 pmol/m$^2$/s of monoterpenes) were emitted in the K17-meanFlx case. The model was initialized with the steady state VOC in-cloud mixing ratios reached by UCLALES-SALSA with the mean fluxes - 1 ng/kg of monoterpenes and 2 ng/kg of isoprene. The mean VOC air mixing ratios measured during the HiWinGS cruise were substantially higher - 85 and 23.5 ng/kg of monoterpenes and isoprene respectively - than the ones reached by the model

with the mean fluxes. Especially for monoterpenes, this imbalance between the marine fluxes and air concentrations was also noticed by Kim et al (2017). For monoterpenes the occasionally observed high peaks also caused the mean to be much higher than the median (15 ng/kg). In order to investigate the impact of such background concentrations, we run additional simulations where the model was initialized with both of those mixing ratios (K17-meanFlx-meanInic and K17-meanFlx-medInic).

As the VOC fluxes also exhibited high spatial and temporal variability and values much higher than the mean were regularly observed during the campaign, an extra simulation (K17-maxFlx) was run with the reported maximum fluxes, which were more than an order of magnitude larger than the mean (18 pmol/$m^2$/s of isoprene and 27 pmol/$m^2$/s of monoterpenes). As these high emissions are not expected to be long-lasting, the model was initialized with the steady state conditions of 1 and 2 ng/kg to see the impact of short-term emission peaks.

To study the impact of the selected oxidant concentrations, the simulations were repeated with doubled ozone concentration and OH and NO3 increased by an order of magnitude, representative of areas with higher anthropogenic impact, such as Northern Atlantic (indicated with -hiOx in Table 2 and case names).

As no sea spray was emitted in these simulations, we evaluate the effects of the emissions by comparing with the NoEms case. As seen from Figure 6 and Figure 7, SOA formed from marine monoterpenes and isoprene can have as strong impact
on clouds as the direct sea spray emissions at 10 m/s windspeed. However, the K17-meanFlx case (green line) with the mean fluxes and steady state background VOC concentrations does not differ from the case without VOC emission (grey) noticeably even with the higher oxidant levels. As the mean monoterpene air concentration observed by Kim et al. (2017) was influenced by the rare but high peak values, we ran simulations initializing the model also with their median observed mixing ratio, which we considered more representative of an average day. The simulation with low oxidant concentrations representing clean marine atmosphere (light blue line on Figure 6) also does not differ from the NoEms case in a noticeable
manner, while some impact can be observed in more polluted atmosphere (blue line on Figure 7). The mean initial concentration (yellow) has the largest effect, but the maximum fluxes (light and dark red on figures 6 and 7) can also slow down the transition to open cell structures by hours even with low initial concentration. The impact is dependent on the rate of oxidation that is defined in the model by prescribed oxidant concentrations.

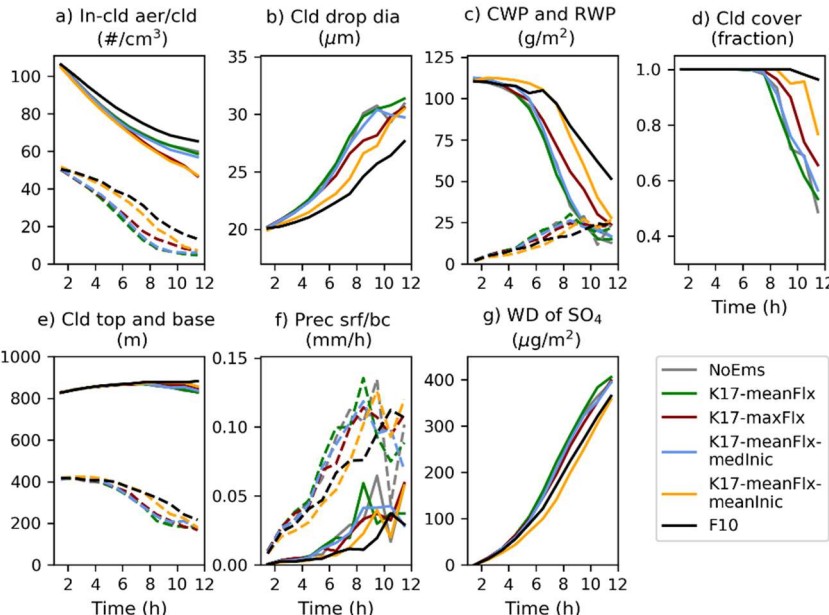


**Figure 6. Impact of VOC emissions. Panels are the same as Figure 3. Gray – no emission,; green – mean fluxes of monoterpenes and isoprene from Kim et al (2017), dark red - maximum fluxes, blue – initialized with observed median concentrations, mean fluxes emitted, yellow – initialized with mean observed concentrations, mean fluxes emitted; black – base (F10 sea spray, no VOC) for comparison.**

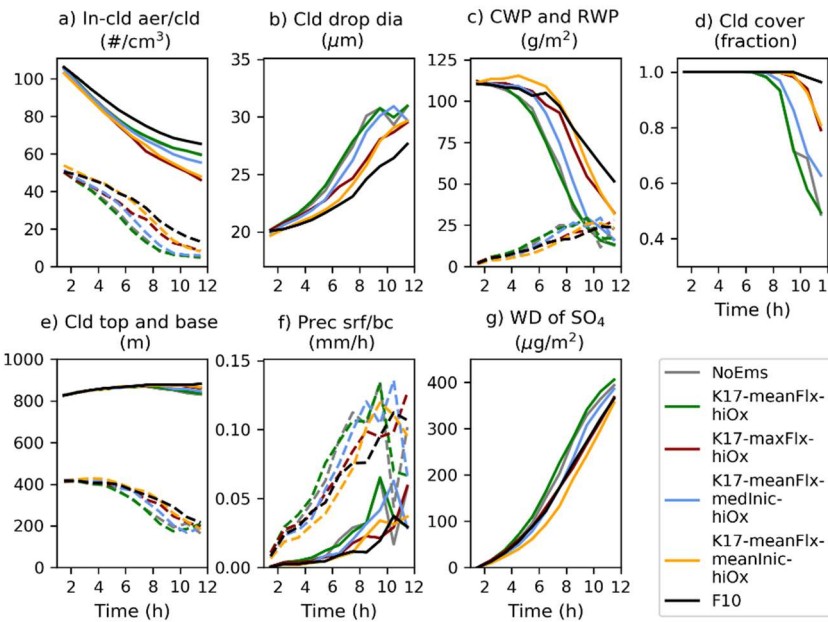


**Figure 7. Impact of VOC emissions in high oxidant case. Panels are the same as Figure 3. Gray – no emission; green – mean fluxes of monoterpenes and isoprene from Kim et al (2017), dark red - maximum fluxes, blue – initialized with observed median concentrations, mean fluxes emitted, yellow – initialized with mean observed concentrations, mean fluxes emitted; black – base (F10 sea spray, no VOC) for comparison.**

In order to elucidate the individual contributions from isoprene and monoterpenes, two simulations were made based on the K17-maxFlx -hiOx simulation by zeroing either isoprene or monoterpene emission and emitting the maximum observed flux of the other VOC (K17- onlyIsop with only isoprene and K17- onlyMtrp with only monoterpenes emissions). As seen from the sensitivity studies where either isoprene or monoterpene emissions were zeroed (green and pink on Figure 8), the whole SOA production in the model and its effect to clouds was solely due to monoterpenes.

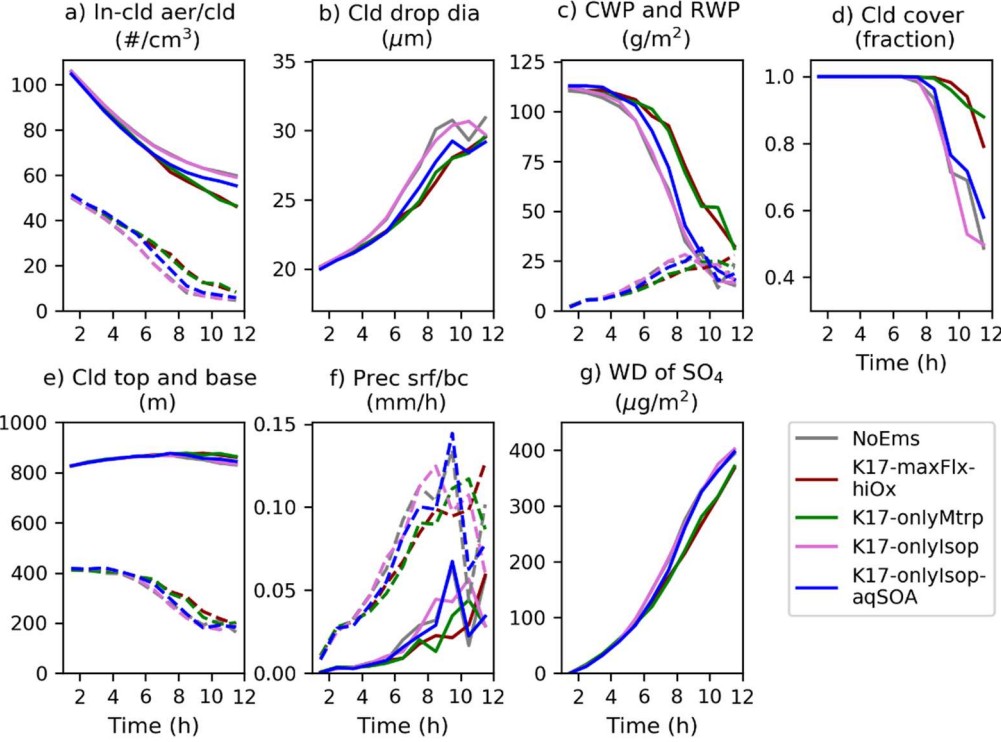

**Figure 8. Contributions from monoterpenes and isoprene and impact of including aqueous SOA formation. Gray – no emission, dark red - maximum fluxes of monoterpenes and isoprene emitted, higher oxidant levels; green – only monoterpenes emitted, pink – only isoprene emitted, blue – same as pink but aqueous formation of SOA included.**

To save the computational resources, the aqueous phase SOA formation from isoprene had been turned off in all the previously discussed simulations. The impact of these processes was investigated with the K17-onlyIsop-aqSOA setup which was otherwise identical to K17-onlyIsop but accounted also for aqueous phase SOA formation through isoprene epoxydiols and glyoxal. Isoprene products indeed became visible when the aqueous phase production was turned on (blue line on Figure 8). However, their impact still stayed significantly lower than that of monoterpenes.

**3.5 Sensitivity to background aerosol size distribution**

In the high VOC emission case (K17-maxFlx-hiOx, dark red on Figure 6) the transitioning timescale to open cell structure is extended by ~3 hours. As the mass fraction of SOA stays very small (<2.5% within 1 km height from surface), its unexpectedly large impact requires further investigation. The background aerosol is defined by Ackerman et al (2009) as bimodal distribution of ammonium bisulfate with high number concentration of Aitken mode aerosol below the sizes that readily act as CCN  (Figure 9). Comparing the simulations with and without marine VOC emissions (Figure 10, left panel) shows that the condensing organic vapours help these particles to grow to the next size bin, though even then they mostly stay too small to contribute much to the activated CCN (Figure 10, right panel). However, both the aerosol and the cloud droplet concentration of the coarser bins stay higher in the simulation with the SOA formation indicating that the large impact stems from positive feedback of the extra particles delaying the drizzle that would otherwise scavenge larger fraction of the particles from the whole size range.

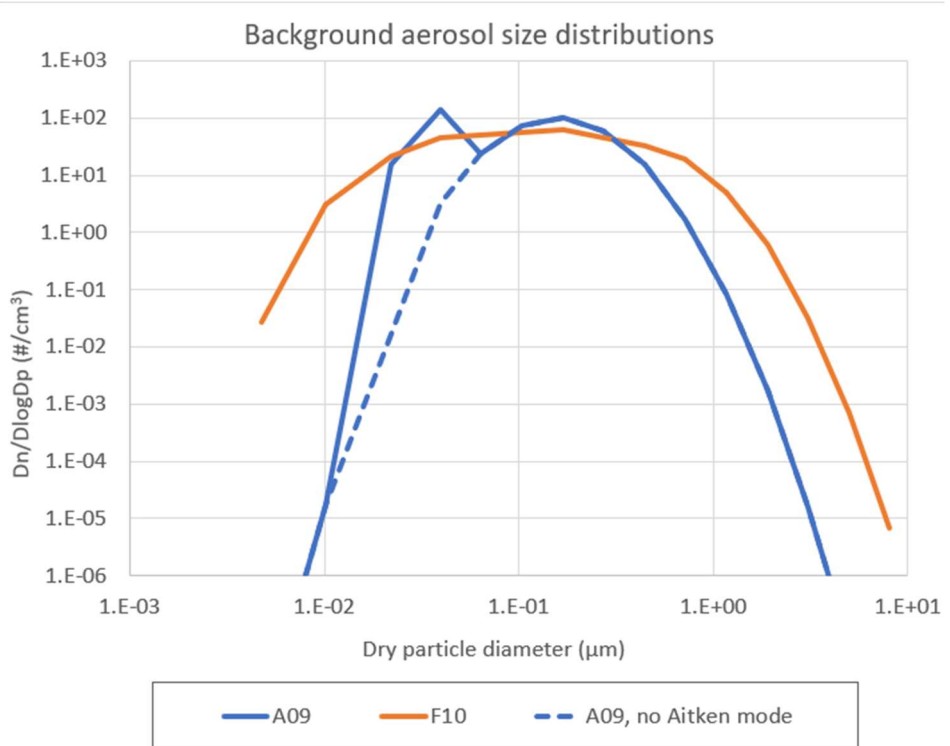

**Figure 9. Model initial condition background aerosol size distributions. Solid blue – bimodal ammonium bisulfate size distribution defined by Ackerman et al (2009), dashed blue – same with fine mode removed, orange – sea salt emission distribution from Fuentes et al (2010).**

Considering this we performed two sensitivity studies – one with removing the fine size mode entirely (K17-maxFlx-aerInic=cr) and another initializing the model with sea salt aerosol with size distribution defined by (Fuentes et al., 2010) emission scheme (K17-maxFlx-aerInic=sslt) as background (see Figure 9). In the latter case the concentration was

normalized to reproduce the same number of cloud droplets as the noEms case immediately after the model spin-up. The
setups were otherwise identical to K17-maxFlx-hiOx simulation. Corresponding no-emission control simulations without
any VOCs were run for both of the alternative initial conditions (NoEms-aerInic=crsand NoEms-aerInic=sslt). The results of
these simulations are shown on Figure 11.

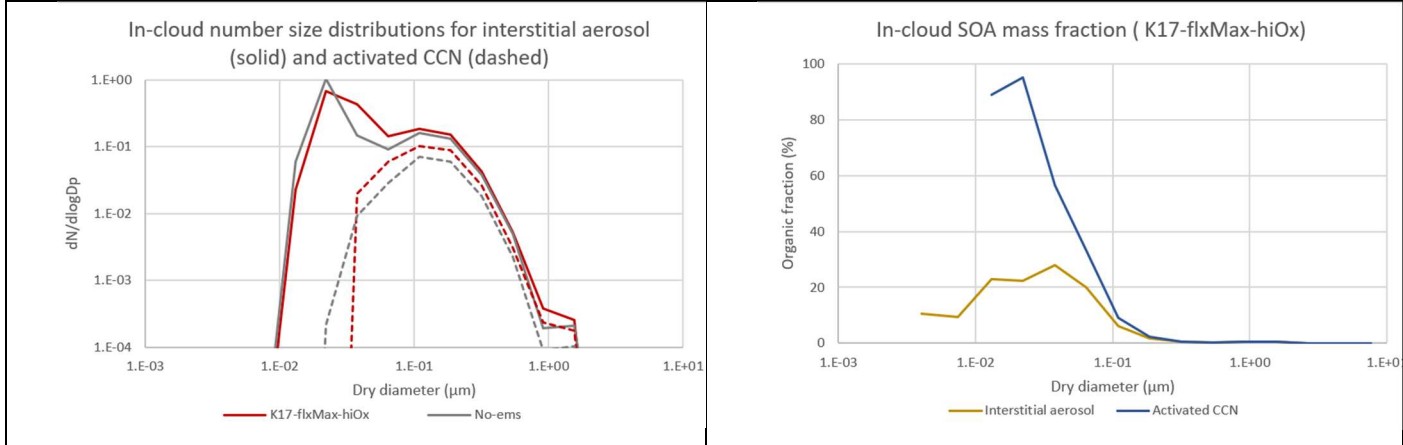

**Figure 10. Left – in-cloud total particle (aerosol + cloud, solid lines) and cloud droplet (dashed lines) dry-size number**
**distribution, right – in-cloud SOA mass fraction in interstitial aerosol (beige) and activated CCN (blue) for the K17-maxFlx-hiOx**
**case, 6 hours after simulation start, mean over cloud layer.**

Comparing the three emission free control simulations (grey, pink and cyan lines on Figure 11) shows that the reduced
initial aerosol number concentration itself leads to slightly shorter lifetime of the closed cell regime in both of the alternative
background cases although both of the alternative initial conditions reproduce the same cloud droplet number in the
beginning of the simulations (dashed lines on panel a). Comparing the pink line with the red one shows that the effect of
VOC emission disappears completely when the Aitken mode is removed from the background aerosol. Thus, we conclude
that the large VOC effect critically depends on the underlying background aerosol distribution. Marine aerosol regularly
exhibits a bimodal distribution (Hoppel and Frick, 1990; Hudson et al., 2015), where the dip between the modes is caused by
activation of large enough particles and their subsequent growth due to cloud processing or removal through wet deposition.
Both in-cloud coalescence and wet deposition of activated particles lead to reduction of particles available to act as CCN.
The small mode consists of primary and secondary particles that are too small for cloud activation in their current size but
capable of growing to CCN size by condensation of organic vapours. Larger number of small particles are likely to exist
downwind marine areas with high biological activity as a result of DMS emission, oxidation and following aerosol
nucleation (Sanchez et al., 2018). However, as seen from Figure 1, some of the sea spray parameterizations also produce
noticeable number of particles in this size range, although they are not separated from the larger particles by a gap. Thus, it is
interesting to consider the case where the background aerosol consists of freshly emitted sea spray. Comparing the
simulations without and with VOC emission (cyan and teal lines on Figure 11) shows that condensation of the organic
vapours on the wide fresh sea salt distribution from Fuentes et al (2010) also delays the transition to open cell regime,

although in lesser extent. Thus, significant impact of marine VOCs on stability of the cloud layer is more likely in areas with high biological activity, as they can provide both larger amount of Aitken mode particles and the organic vapours to condense on them to grow them to CCN size.

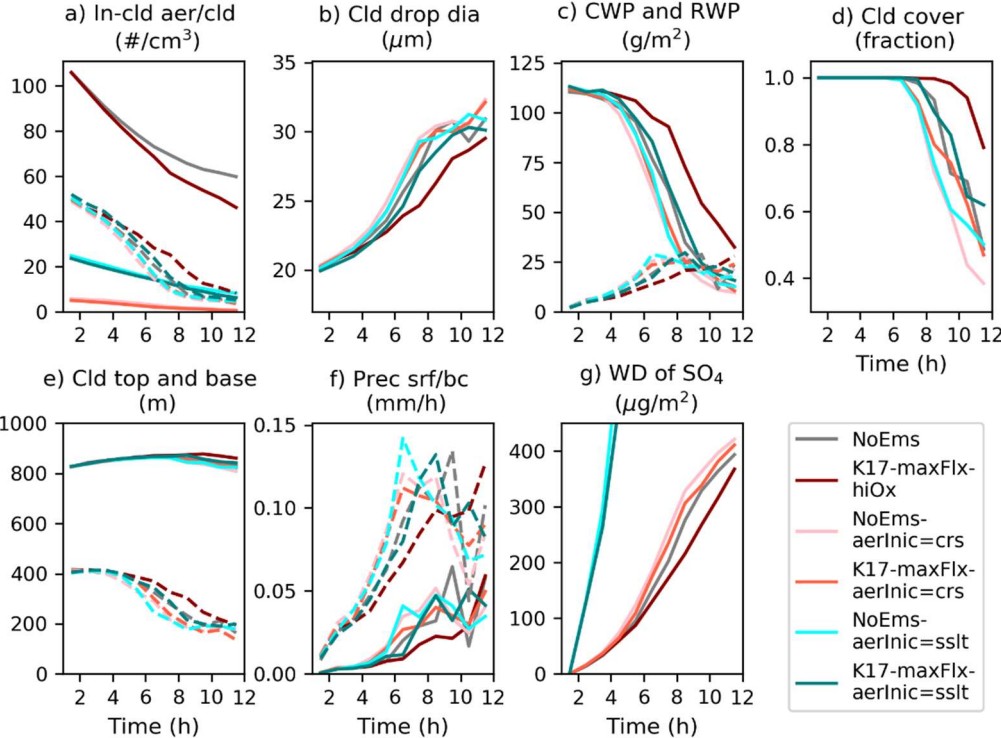

**Figure 11. Impact of background aerosol on VOC effect. Dark red (K17-maxFlx-hiOx) - maximum fluxes of monoterpenes and isoprene emitted, higher oxidant levels, background aerosol initialized with Ackerman et al (2009) size distribution, Gray – no-emission control case for K17-maxFlx-hiOx,; red (K17-maxFlx-aerInic=crs) – VOCs same as K17-maxFlx-hiOx, fine mode removed from the initial background aerosol, pink – no-emission control for K17-maxFlx-aerInic=crs, teal (K17-maxFlx-aerInic=sslt) – VOCs same as K17-maxFlx-hiOx, background aerosol initialized with sea salt size distribution from Fuentes et al (2010), cyan – no-emission control for K17-maxFlx-aerInic=sslt.**

## 4    Discussion

Compared to differences between different sea spray emission parameterizations and their temperature dependences, we found the sensitivity of the clouds to all other uncertainties in the marine emissions to be relatively limited. According to the review by (Grythe et al., 2014), different parameterizations agree better in both mass and number fluxes in the size range between ~0.1 and 2 μm but tend to diverge in both finer and coarser direction. All these parameterizations can agree equally well with observations of aerosol optical depth (AOD) at 550 nm that is most sensitive to particles within this size range, or sea salt mass observed in stations located long distance from the emission region as the largest particles have short

atmospheric lifetimes. However, as seen from the Figure 11, panel b, sea salt particles smaller than 0.1 μm can make up a noticeable fraction (~20%) of the activated cloud droplets. While fine sea spray provides extra cloud condensation nuclei and delays the onset of drizzle as the collision-coalescence process is slowed down due to smaller cloud droplet mean size, the coarse mode has an opposite effect due to GCCN speeding up the drizzle formation through the enhanced collision-coalescence processes. Threshold diameters between 2 and 10 microns have been used in previous literature to define the sea spray GCCN (Dror et al., 2020). In our simulations we have not investigated where exactly the threshold would be that would distinguish the different effects, but we do see opposite impacts from emitting submicron or supermicron sea spray. The balance between the CCN and GCCN impacts depends on the size distribution of the employed emission parameterization and according to our results the large differences in those can change even the direction of how the sea spray affects the stability of the cloud layer.

As the current study required a large number of simulations and the GCCN effects were relevant for only small fraction of those, we used the Seifert-Beheng autoconversion parameterization to reduce computational burden. However, as this scheme is based on total droplet number and mass and thus does not resolve their size spectrum, it might not be optimal for modelling the effect of the GCCN. In the Supplementary Material Section 1.3 we show simulations that employ an alternative, more mechanistic scheme for drizzle formation. While the onset of drizzle in those simulations happens much later due to the long time required for the scheme to build up realistic droplet size distribution, the main conclusions regarding the fine mode particles stay the same. However, the impact of GCCN is noticeably enhanced, indicating that their role might be even more important than shown above. One important reason for the difference between the two schemes is that with the Seifert-Beheng scheme in the no-emission case the precipitation starts relatively early in the simulation. The coarse sea spray particles emitted from the sea surface since the beginning of the simulation have simply not yet reached the cloud level in noticeable amounts. The long time required for the collision-based scheme to reach realistic cloud droplet size distribution allows also more GCCN to reach cloud level and influence the development of precipitation.

Another place where GCCN play a role is the case of organic fraction (Figure 5) and there reducing the hygroscopicity of the coarse fraction has a noticeable impact as more sea spray has had time to reach cloud level. The difference caused by the size-dependence of sea spray organic fraction is noteworthy. Attributing a large organic fraction to the coarse sea spray lowers its water uptake and substantially reduces the GCCN effect keeping the cloud field stable noticeably longer. As observations do not support large organic fraction in coarse sea spray, caution is needed when using organic fraction parameterizations that do not include size dependence.

It is noteworthy that significant effects from SOA to the cloud layer are seen only when emitting the largest reported VOC fluxes or initializing the model with the mean observed mixing ratios that were far higher than the steady state mixing ratios reached with the mean observed fluxes. Box model calculations by Kim et al. (2017) also show that the fluxes necessary to sustain the atmospheric mean monoterpenes mixing ratios observed by them or by Yassaa et al. (2008) noticeably exceed their observed mean fluxes. Yassaa et al. (2008) presents compelling evidence for oceanic origin for their observations making it unlikely that this mismatch would be solely due to high contribution from non-oceanic sources. Kim

et al. (2017) suggest higher contribution of lower reactivity monoterpenes as an alternative explanation for the imbalance. While this could be the case for their total monoterpenes measurements, the observations of Yassaa et al. (2008) identified α-pinene as the major contributor to their measurements. Although large uncertainties are introduced by the simple treatment of the oxidants in our model, according to our results the equilibrium mixing ratio reached by the highest fluxes is not so far from the observations when taking into account the vertical profiles of the surface emitted VOCs. As seen in Figure 12, depending on the oxidant level, the surface and cloud level concentrations can differ from a factor of two up to more than an order of magnitude. The observations of Kim et al (2017) (85 ng/kg mean and 15 ng/kg median for monoterpenes) were made at the altitude of 17 meters. The high observed concentrations at this height could be at least partly explained if monoterpene fluxes comparable to the maximum observed ones would have been relatively common in the upwind areas during the measurement campaign.

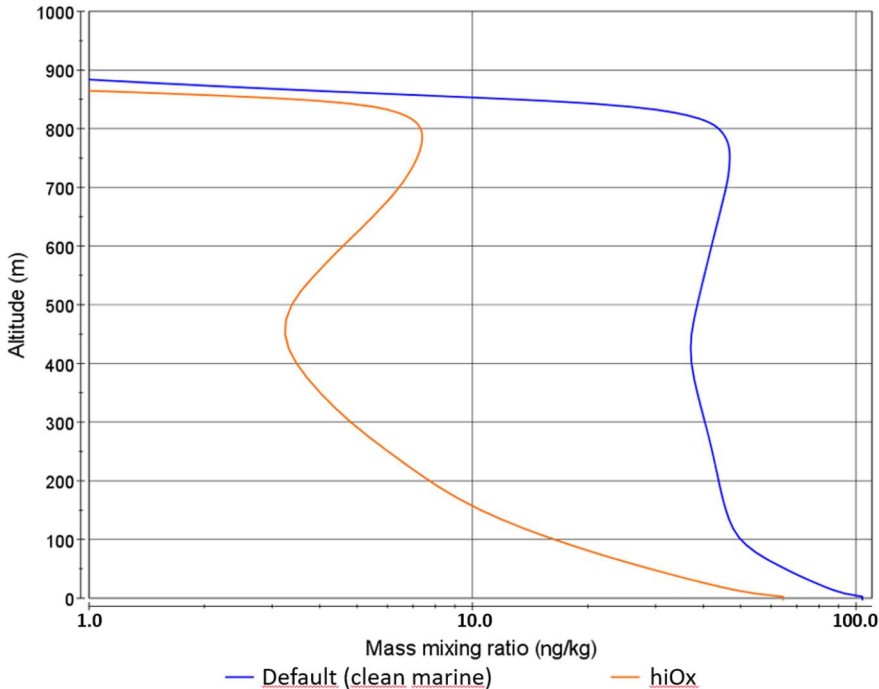

**Figure 12. Vertical profiles of equilibrium monoterpene mass mixing ratios rising from the maximum flux observed by Kim et al., (2017) with higher and lower oxidant concentrations.**

Although we did not detect any immediate effect from the gas phase isoprene oxidation for the conditions of the DYCOMS II campaign, it is possible that it would become more noticeable in longer time scales or in different meteorological conditions than in the simulated case. The simulations of Gantt et al. (2009) with 3% aerosol yield from isoprene also showed only very minor impact of isoprene derived SOA to total marine organic aerosol as global annual average, but reported much higher short term contributions over the tropical regions when high isoprene emissions coincided with small primary organic fluxes due to their different dependencies on meteorological drivers such as windspeed. Similar

results were shown by Prank et al (2018) for the biogenic organic aerosol in Europe using a similar VBS scheme without aqueous phase processes – the majority of SOA in the model originated from vegetation emitted monoterpenes while the role of isoprene was modest. The isoprene SOA yields used in UCLALES-SALSA are on the higher end of what has been reported in the literature (0.0295 for the VBS bin with saturation concentration of 1 $\mu g/m^3$ and 0.0453 for 10 $\mu g$ bin/$m^3$), suggesting that at least in the conditions and timescale of the study, gas phase production of semivolatile organics from isoprene oxidation does not lead to enough low volatility products to have a noticeable impact. However, significant amounts of isoprene oxidation products have been observed in remote marine aerosol by e.g. Hu et al., (2013). As pointed out by Yu and Li (2021), the role of isoprene in forming marine aerosol could be underestimated in current models as substantially larger SOA yields have been shown using more sophisticated chemistry schemes (e.g. Bates and Jacob 2019), especially when accounting for aqueous phase chemical processes involving isoprene epoxydiols (Nguyen et al., 2014). This assessment is confirmed by our simulations - isoprene products became visible when the aqueous phase SOA production is turned on, although their impact is still significantly lower than that of monoterpenes. Use of a more comprehensive chemistry scheme can also lead to reduction of SOA, as chemical interactions between isoprene and monoterpene oxidation products can reduce the yield of low volatility compounds from monoterpenes (McFiggans et al., 2019). Unfortunately, a sophisticated gas phase chemistry scheme is currently not feasible in a LES model like UCLALES-SALSA due to prohibitively high computational requirements.

In order to ensure direct comparability of the magnitudes of the studied effects, we applied the different emission manipulations one by one. However, we expect the cloud impacts caused by marine emissions through different pathways to be very close to additive. For instance, some of our preliminary simulations (not shown) demonstrated that the effects of the additional CCN from the Fuentes et al. (2010) parameterization and the reduced hygroscopicity due to the organic fraction mostly cancel each other out. Also, the changes caused by the dissolved organic carbon in the sea surface layer to the emission flux and the hygroscopicity of the emitted particles both depend on the composition of the DOC or the type of the algal exudate (Fuentes et al., 2010, 2011). On the other hand, the effects of SOA formation would be smaller when including sea spray emission as in higher aerosol load the semivolatiles would have larger number of particles to condense on and thus individual particles would grow less. However, the gaseous VOC and particulate sea spray emissions have somewhat different drivers. The VOC emission depends mostly on the production of the species by the phytoplankton. The sea spray emission has much stronger dependence on windspeed, and thus its emission can be minuscule in quieter conditions.

In the current study we concentrated on how the marine emissions affect the properties of lightly drizzling stratocumulus. As the sensitivity of marine stratocumulus to aerosol and precursor emissions depends for instance on background aerosol, cloud thickness and drizzle intensity (Terai et al., 2012; Wang et al., 2011), the results could differ for different meteorological conditions. Grythe et al (2014) reports that the maximum annual sea spray production occurs with winds in the range of 7-16 m/s for the emission schemes they reviewed. We selected 10 m/s as a representative windspeed of this range to force the sea spray emission schemes in our simulations. However, the most frequent wind speeds over the ocean are somewhat lower, between 5 and 7 m/s (Grythe et al., 2014). In these conditions the sea spray effects would be

substantially smaller as the fraction of sea surface covered with whitecaps where the bubble mediated sea spray production takes place depends strongly on the wind speed. For example, 10-meter wind is in power 3.41 in the widely used whitecap parameterization of Monahan et al (1986). This also means that only about 30% higher windspeed would be needed to balance the loss of CCN due to in-cloud coalescence that in the base case exceeds the emission rate by ~2.5 times.

**4     Conclusions**

In this study we used the UCLALES-SALSA large eddy simulator that is coupled to a cloud microphysical model that includes a detailed description of aerosols, clouds and precipitation to model the conditions of the DYCOMS-II observational campaign characterized by low level stratocumulus clouds transitioning from closed cells to drizzling open cell structure. Our aim was to investigate the impacts of sea spray and marine VOCs on cloud stability. Our simulations were 590     designed to cover the ranges of the driving parameters (SST, DOC, Chl α, and terpenoid fluxes) that would have the largest impact.

Including a sea spray emission scheme with high flux of Aitken and accumulation mode particles, such as parameterized by Mårtensson et al (2003) or Fuentes et al (2010) to our simulations had a noticeable impact, delaying the transition to open cells by several hours by delaying the onset of drizzle, as the extra CCN lead to activation of larger number of cloud droplets 595     and slower growth of droplet size. Opposite effect was seen from coarse particles acting as giant CCN that speed up the coalescence of cloud droplets and drizzle formation.

While the importance of sea spray emission temperature dependence has been pointed out by several sources (Grythe et al., 2014; Liu et al., 2021; Mårtensson et al., 2003), its magnitude and even direction differs between different parameterizations. For our selected parameterizations the effects of SST varied from negligible in case of the Forestieri et al. 600     (2018) parameterization to closed cell regime lasting beyond the experiment timescale for the commonly used Mårtensson et al. (2003) scheme. Varying the sea water organic content from zero to the high limit of our considered parameterizations had smaller impact on the clouds than would be seen from changing SST between summer and winter conditions in the Mårtensson et al (2003) parameterization. However, it was important to account for the size dependence of the organic fraction of the emitted particles, as changing the hygroscopicity of the coarse fraction has opposite effect to that of fine. 605     Previous studies have shown that while marine stratocumulus can be highly sensitive to giant CCN, the related uncertainties are large due to complex dependencies on the background CCN concentration (Feingold et al., 1999) and GCCN size distribution (Dror et al., 2020). Our results show that the effect of GCCN is also highly sensitive to the assumptions regarding the water uptake of these coarse particles, making the size dependence of sea spray organic fraction relevant for models with comprehensive autoconversion schemes that account for the GCCN effects.

According to our results, the secondary organic aerosol formation from terpenoids can have a large impact, delaying the drizzle and extending the lifetime of the closed cell regime up to several hours. The SOA effect critically depends on the background aerosol distribution – condensation of semivolatiles can affect the clouds when sufficient Aitken mode particles

are present to grow into CCN size. Nucleation of oxidation products of DMS emitted by marine phytoplankton is considered as an important source of new particles in clean marine atmosphere. As the emissions of DMS and other VOCs coincide in the biologically active areas, the terpenoid emissions could potentially play important role in growing the newly formed particles to CCN size.

In the considered conditions and timescale, gas phase production of semivolatile organics from isoprene oxidation did not lead to enough low volatility products to have a noticeable impact even with the highest isoprene levels. Simulating the reactive partitioning IEPOX and glyoxal to aerosol water, on the other hand, had a noticeable impact, stressing the importance of including in models the aqueous phase SOA production processes.

Significant effects from monoterpenes were seen in some simulations, extending the transitioning timescale from closed to open cell structure by hours. However, cloud stability was noticeably impacted only when emitting the largest VOC fluxes reported in literature or initializing the model with the mean observed mixing ratios that were far higher than the steady state reached with the mean observed fluxes. More observations of monoterpenes in ocean water and air above it are needed to understand the frequency of such high fluxes and role of these species in marine atmosphere.

## 5    Code and data availability

The simulation data presented in this paper together with the source code of the version of UCLALES-SALSA used for creating the data are available from https://fmi.b2share.csc.fi/records/9f230bd7553a40aeadafd400cedeba52 (Prank et al., 2022)

## 6    Author contributions

MP and TR designed the study. MP performed and analysed the model simulations. All authors have contributed to developing the UCLALES-SALSA model. MP prepared the manuscript with contributions from all co-authors.

## 7    Competing interests

The authors declare that they have no conflict of interest.

## 8    Acknowledgements

This study was funded by the Academy of Finland projects 322532 and 317390 and FORCeS project of Horizon 2020 (grant no. 821205). Computational resources were provided by CSC – IT Center for Science, Finland.

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
