# Peer review of "Impacts of marine organic emissions on low level stratiform clouds – a large eddy simulator study"

_Atmospheric Chemistry and Physics, 2022_

## Referee Comment (RC1)

**Review of "Impacts of marine organic emissions on low level stratiform clouds – a large eddy simulator study" by Prank et al. (acp-2022-265)**

The manuscript addresses how aerosol particles formed from sea spray or marine gas-phase emissions affect the colloidal stability of stratocumulus clouds using large-eddy simulations. The authors compare different parameterizations of the aforementioned aerosol sources, showing that their effects on stratocumulus stability can vary substantially based on the type of parameterization applied in their simulations. While I generally appreciate the authors' efforts and agree with most results, I have substantial concerns about how well the effects of giant cloud condensation nuclei (GCCN) on the collision/coalescence process are represented by the applied cloud microphysics scheme. As the effects of GCCN constitute one pillar on which the study is based, I cannot support the publication of this study in its current form. Please note that my review focuses on cloud microphysical processes and their modeling, as my expertise does not cover the chemical details addressed in other parts of the study.

**Major Comments**

*Description of the cloud microphysical model.* While the authors spend several lines on how the different chemical species and types of aerosol are considered in their model (ll. 118 – 181), I miss a similarly detailed explanation on the applied cloud microphysical model. There are some hints on how cloud microphysics are represented in Sec. 2.2, but I miss a more concise description. The most important questions are: How is the aerosol hygroscopicity considered in the condensational growth of droplets? Is condensational growth explicitly represented or treated by a saturation adjustment scheme, as often done in simpler cloud microphysical models? Do the authors account for supersaturation changes on a sub-timestep level (e.g., Clark 1973)? How does Seifert and Beheng (2001) autoconversion scheme consider GCCN? These questions are crucial because the way these processes are represented in the cloud microphysical model affect the simulated cloud and its behavior.

*The effect of GCCN (ll. 270 – 273).* As outlined above, I doubt that the applied collision/coalescence scheme can adequately represent the effect of GCCN, as it is only based on the number and mass of cloud and rain drops, without considering the GCCN explicitly. While I might miss a detail here, the relatively minuscule effects of the GCCN shown in Fig. 3c (blue and gray lines) do not indicate a substantial effect. While it is known that precipitating clouds are not very susceptible to the addition of GCCN, the results agree with my expectation, but this behavior might also be caused by an insufficiently represented GCCN effect on collisional growth.

**Minor Comments**

L. 27: Define SOA.

Ll. 8 – 29: The abstract is too long. Consider shortening it.

L. 12: A large-eddy simulation model is used to simulate dynamics, not aerosol particles, cloud droplets, or rain drops. A cloud microphysical model does this.

Ll. 17 ff.: The concept of a lifetime for stratocumulus clouds is odd. I would rather refer to the transitioning timescale between closed and open cells.

Ll. 38 – 39: The role of longwave radiative cooling in causing and sustaining a cloud, especially a stratocumulus cloud, should not be understated.

L. 40: The aerosol size distribution contains information on the aerosol number concentration. I would rather write about aerosol number concentration and aerosol size as the controlling factors.

Ll. 124 ff.: Here and other places: Do these values refer to particle radius or diameter?

Ll. 191 – 192: Open stratocumulus cells can be much larger than 10 km (e.g., Kazil et al. 2017).

Ll. 255 – 258: How is the droplet size calculated? From the number concentration and mass of the droplets?

Fig. 11: Is it possible to show the aerosol and droplet size distributions not as a bar graph? It is very hard to decern differences between the simulated cases.

**Technical Comments**

Please indent the first line of a new paragraph, as it is done in almost all publications. Without this optical help, it is very hard to read the text.

If aerosol particles are addressed, please write "aerosol particles" and not "aerosols", as the latter may also refer to different species of aerosol.

L. 61: Citation style.

L. 84: "volatile VOCs" is tautological.

**References**

Kazil, J., Yamaguchi, T. and Feingold, G., 2017. Mesoscale organization, entrainment, and the properties of a closed-cell stratocumulus cloud. *Journal of Advances in Modeling Earth Systems, 9*(5), pp.2214-2229.

Clark, T.L., 1973. Numerical modeling of the dynamics and microphysics of warm cumulus convection. *Journal of the Atmospheric Sciences*, *30*(5), pp.857-878.

Seifert, A. and Beheng, K.D., 2001. A double-moment parameterization for simulating autoconversion, accretion and selfcollection. *Atmospheric research*, *59*, pp.265-281.

---

## Author Response (AR1)

**Response to reviewers**

We thank the reviewers for their comments. Our answers are given below in blue. To answer the concerns of the reviewers we have extended the model description in the manuscript and included Supplementary Information with sensitivity studies to model resolution, model noise and autoconversion parameterization.

**Anonymous Referee #1**

**Major Comments**

Description of the cloud microphysical model. While the authors spend several lines on how the different chemical species and types of aerosol are considered in their model (ll. 118 – 181), I miss a similarly detailed explanation on the applied cloud microphysical model. There are some hints on how cloud microphysics are represented in Sec. 2.2, but I miss a more concise description. The most important questions are: How is the aerosol hygroscopicity considered in the condensational growth of droplets? Is condensational growth explicitly represented or treated by a saturation adjustment scheme, as often done in simpler cloud microphysical models?

The model description has been amended with more details on microphysics. Condensational growth is represented explicitly and hygroscopicity is taken into account according to the Köhler theory depending on the composition and size of particles in every size bin.

Do the authors account fo supersaturation changes on a sub-timestep level (e.g., Clark 1973)?

Clark (1973) showed large differences between simulations with 10 and 5 second timestep. The differences between their simulations with 5 and 2.5 second timesteps were already significantly smaller. The maximum length of timestep in our simulations is 1 second and, if needed, it is automatically reduced during the simulation to maintain the CFL criteria in the transport processes. Within a single model timestep, the condensation equation and thus the water partitioning between gas and liquid phases is solved in even shorter subtimesteps (0.05s) to avoid problems with small aerosol particles that quickly respond to changes in conditions. Overall, while we are not using sub-timestep adjustments for total water content in a grid point, we are not expecting this to cause major errors in the results.

Description of the sub-stepping has been added to the model description.

How does Seifert and Behen (2001) autoconversion scheme consider GCCN? These questions are crucial because the way these processes are represented in the cloud microphysical model affect the simulated cloud and its behavior.

The scheme developed by Seifert and Beheng (2001) indeed only uses the total droplet number and mass and thus does not resolve their size spectrum to the level needed to accurately reproduce the GCCN effect. However, the currently used bin scheme describes the dry particle size distribution accurately, which is required for modelling the GCCN effect. We explicitly model the aerosol and cloud droplet growth through condensation and coagulation-coalescence processes for every size bin (including the GCCN bins) which are then summed to obtain the total mass and thus, the effect is at least partially present in the simulations. The autoconversion scheme moves cloud droplets from dry-size based cloud bins to droplet-size based rain bins, which increases the accuracy of modelling

collision/coalescence processes between droplets in cloud and rain bins. All microphysical processes are simulated otherwise identically for aerosol, rain and cloud droplets.

These details have been added to the model description.

The effect of GCCN (ll. 270 – 273). As outlined above, I doubt that the applied collision/coalescence scheme can adequately represent the effect of GCCN, as it is only based on the number and mass of cloud and rain drops, without considering the GCCN explicitly. While I might miss a detail here, the relatively minuscule effects of the GCCN shown in Fig. 3c (blue and gray lines) do not indicate a substantial effect. While it is known that precipitating clouds are not very susceptible to the addition of GCCN, the results agree with my expectation, but this behavior might also be caused by an insufficiently represented GCCN effect on collisional growth.

The reviewer is correct in stating that the Seifert-Beheng scheme is not optimal for modelling the giant CCN effect. Schemes that track more accurately the growth of individual droplets, such as the super-droplet schemes (Shima et al., 2009), could be more accurate. However, also the super-droplet schemes have limitations such as high computational costs that would make them impractical for the current study.

However, UCLALES-SALSA includes an alternative, more mechanistic option for simulating droplet growth from cloud droplet to small drizzling droplet. This scheme has been used previously for studying cloud seeding (Tonttila et al., 2021) and in an aerosol-cloud closure study (Calderón et al., 2022). It is based on explicitly counting all collisions between cloud droplets where the product is large enough (>20 microns) to start efficiently collecting other droplets to form drizzle and rain. This collision rate limited scheme requires much longer spin-up to build up realistic cloud and drizzle droplet size distributions than employing an autoconversion parameterization that grows a subpopulation of cloud droplets instantly to sizes above 50 micrometers. As the current study required a large number of simulations and the GCCN effects were relevant for only small fraction of those, we used the Seifert-Beheng parameterization to reduce the computational burden. We have now repeated the three simulations mentioned by the reviewer using the mechanistic scheme. Figures 1 and 2 below show the simulations using the Seifert-Beheng parameterization and the mechanistic scheme, respectively. Figure 1 indeed shows relatively minuscule effects of the GCCN. One important reason for this is that with this scheme the precipitation starts relatively early in the in the no-emission case. The sea spray particles emitted from the sea surface during the simulation have not yet affected the clouds noticeably, and there are very few coarse particles present in cloud level. Another place where the GCCN are mentioned in the manuscript is the case of organic fraction (Figure 5 in the main paper), and there reducing the hygroscopicity of the coarse fraction has a slightly larger impact as more sea spray has had time to reach cloud level.

As seen from Figure 2, with the mechanistic scheme it takes about twice longer in the no-emission case (gray line) before the clouds start restructuring and significant levels of precipitation occur. And in this case the effect of the GCCN emitted by the Gong (2003) sea spray scheme (blue line) is indeed much more noticeable, speeding up the appearance of surface precipitation by several hours. An indication of the GCCN effect is also visible for the F10 case (black) where low levels of below-cloud drizzle are visible hours before the precipitation occurs in the no-emission case. In this case the GCCN effect competes with the rain-delaying effects of the extra CCN from fine sea spray and thus stronger surface-reaching drizzle still starts a couple of hours later than it occurs in the no-emission control, similarly to the main simulations.

[Figure]

*Figure 1. Simulations with the Seifert-Beheng autoconversion parameterization. Hourly averaged time series, mean over the model area. Panels: a – in-cloud cloud interstitial aerosol (solid) and cloud droplet (dashed) concentration, b – Cloud droplet size, c –Cloud liquid water path (solid) and rain water path (dashed), d – cloud fraction, e - height of cloud top (solid) and base (dashed), f –precipitation rate at surface (solid) and below cloud (dashed), g – cumulative wet deposition of background aerosol (ammonium bisulfate). Simulations: grey – no-emission control, black –F10, Blue – G03 setups. All schemes were run with SST 10°C and 10 m/s windspeed.*

[Figure]

*Figure 2. Simulations using explicit precipitation formation scheme. Panels and line colours are the same as in Figure 1.*

The large difference in the mean cloud droplet size when using the different schemes is due to using different definitions for drizzle – in the case of the Seifert-Beheng scheme the raindrop size spectrum starts from 50 microns and all droplets produced by the autoconversion scheme are assumed to be this size. In the simulations with the mechanistic scheme the drizzle bins are used for coalescence-growth dominated size-range and start from 20 microns. All droplets that reach this size by collision-coalescence process are moved from cloud bins to drizzle bins. Purely condensational growth is still handled in the cloud bins, as condensation affects the whole population in the bin uniformly and the wet-size distribution does not widen. As mentioned above, this classification is not a source of error as all microphysical processes are computed identically for both cloud droplets and drizzle and only the bin limits are defined differently based on dry size (cloud droplets) or wet size (drizzle/precipitation).

The paper has been amended with the discussion of the Seifert-Beheng scheme likely underestimating the GCCN effect and the extra figures comparing the two schemes and related discussion has been included as Supplemental Information.

Minor Comments
L. 27: Define SOA.

Secondary organic aerosol, will be clarified in the paper

Ll. 8 – 29: The abstract is too long. Consider shortening it.

Abstract has been shortened

L. 12: A large-eddy simulation model is used to simulate dynamics, not aerosol particles, cloud droplets, or rain drops. A cloud microphysical model does this.

This will be clarified in the manuscript

Ll. 17 ff.: The concept of a lifetime for stratocumulus clouds is odd. I would rather refer to the transitioning timescale between closed and open cells.

The wording will be changed according to reviewer's suggestion throughout the manuscript

Ll. 38 – 39: The role of longwave radiative cooling in causing and sustaining a cloud, especially a stratocumulus cloud, should not be understated.

The sentence will be amended to include longwave radiative cooling

L. 40: The aerosol size distribution contains information on the aerosol number concentration. I would rather write about aerosol number concentration and aerosol size as the controlling factors.

Will be changed according to reviewer's suggestion

Ll. 124 ff.: Here and other places: Do these values refer to particle radius or diameter?

Here and elsewhere in the paper all sizes are given as dry diameter if not stated differently. This will be made clearer in the revised manuscript.

Ll. 191 – 192: Open stratocumulus cells can be much larger than 10 km (e.g., Kazil et al. 2017).

The reviewer is correct stating that the domain is too small to fully represent the different structures as seen for example in Kazil et al. (2017). In fact, the cells in this case are also larger than 10 km (Figure 3). However, our preliminary tests showed that 10 km domain size was sufficient to simulate the transition process with similar accuracy as the 20km test domain for our purposes. Making all simulations on the larger domain would have severely limited the number of different cases studied.

Explanation of this is added to the paper.

[Figure]

*Figure 3. Cloud and precipitation pattern in comparable simulations with 20 km domain, 50 m resolution and 10 km domain, 60 m resolution. 10 hours after the start of the simulation. Contours – surface precipitation rate (mm/h), white shading –liquid water path (g/m² ) *0.01 (scaled to fit on same scale as precipitation).*

Ll. 255 – 258: How is the droplet size calculated? From the number concentration and mass of the droplets?

Droplet size is calculated from the volume and number of droplets in every size bin and averaged over the cloudy grid cells. Explanation is added to figure caption.

Fig. 11: Is it possible to show the aerosol and droplet size distributions not as a bar graph? It is very hard to decern differences between the simulated cases.

The bar plots have been replaced with line plots for size distributions and organic fraction.

**Technical Comments**

Please indent the first line of a new paragraph, as it is done in almost all publications. Without this optical help, it is very hard to read the text.

If aerosol particles are addressed, please write "aerosol particles" and not "aerosols", as the latter may also refer to different species of aerosol.

L. 61: Citation style.

L. 84: "volatile VOCs" is tautological.

Corrections have been made following reviewer's suggestions

Anonymous Referee #2

**Major concerns**

Many references are not included in the reference list. In particular, references from the Section 1 are missing. Please check for completeness.

We thank the reviewer for noticing this issue. The reference list has been corrected.

All figures are of low quality. Please use more sophisticated plotting routines (e.g., from R, python, …) to adjust line thickness, line color (especially Fig. 1), axis labels, units (many length scales are provided in meters), and appearance of legends.

Figures have been redrawn for better readability. In figure 1 the colors for different temperatures for the M03 parameterization have been made more discernable. Figure legends have been made better readable. Particle sizes on the axes have been changed from meters to micrometers.

In addition, the chosen short names of experiments that are shown in the legend aren't self-explaining and require carefully reading the main text – please improve this also. Accordingly, Table 1 and 2 should be improved.

The reviewer is correct that the short names of the experiments can be unclear. In fact, difficulties in finding short enough self-explaining names for all 26 experiments to include in figure legends were the main reason for listing the experiment setups in Tables 1 and 2 for quick lookup. We have now changed the most confusing experiment names to longer, more self-explaining versions.

Given the range of questions, the paper is structured well. However, individual sections lack clarity and make it hard to follow. I recommend splitting section 3 (that combines results and discussion) into a result section and a discussion section. Having a discussion section may also make it easier to express the advance in knowledge that this study provides.

The Results and Discussion sections have been separated.

The paper could benefit from a proper definition of chemical reactions that are considered in this study. Readers less versed in atmospheric chemistry could benefit from definitions.

A table with reactions has been added to Supplementary Information and the model description has been amended.

Please also clarify: (1) Are oxidants consumed during reactions and are there any (other) sinks or sources for oxidants? (2) Is sunlight necessary for reactions?

No, similarly to Kim et al., (2017), we keep the oxidant concentrations constant at levels representative of longer time averages. That means that they are not consumed, emitted or produced and thus sunlight is not relevant for these simulations.

This in now clarified in the model description.

The study importantly shows how giant CCN affect drizzle but no definition of "giant CCN" is provided. Is there a dry-diameter threshold that the authors use to discern CCN from giant CCN? Would this definition change if a different horizontal/vertical resolution was used?

Threshold diameters between 2 and 10 microns have been used in previous literature to define the sea spray GCCN (Dror et al., 2020). In our simulations we have not investigated where exactly the threshold would be that would distinguish the different effects, but we do see opposite impacts

when emitting only submicron or supermicron sea spray. We are not expecting this to depend on the model resolution given that the cloud pattern is resolved reasonably well.

A paragraph discussing the impact of aerosol particles of different sizes was added to the Discussion section.

Having such a strong focus on microphysical processes, I think it's important to show the dependence on horizontal/vertical resolution as well as domain size. In addition, the authors should produce a small ensemble of simulations for at least one setup (perhaps the baseline) to approximate simulated uncertainty.

In preparation of the simulations presented in the manuscript we performed preliminary simulations with larger domain size (20x20 km) and higher horizontal resolution (50 m). This setup was computationally too heavy to be feasible for the large number of simulations presented, however the current simulations did not significantly differ from those in any of the variables discussed in the manuscript (see Figure 4).

[Figure]

*Figure 4. Timeseries of comparable simulations with 20 km domain, 50 m resolution (brown) and 10 km domain, 60 m resolution (black). Hourly averaged time series, mean over the model area. Panels: a – in-cloud cloud interstitial aerosol (solid) and cloud droplet (dashed) concentration, b – Cloud droplet size, c –Cloud liquid water path (solid) and rain water path (dashed), d – height of cloud top (solid) and base (dashed), e –precipitation rate at surface (solid) and below cloud (dashed), f – cumulative wet deposition of background aerosol (ammonium bisulfate).*

We have not explicitly tested the sensitivity of UCLALES-SALSA model to the vertical resolution for this specific case, as it was optimized for the LES intercomparison study by Ackerman et al., (2009). The resolution is less than 25 meters for all in-cloud and below-cloud layers and 5 meters in the regions with largest gradients (near surface and cloud top).  Stevens et al. (2005) tested the sensitivity of LES models to the resolution at cloud top for the Research flight 1 of the DYCOMS II campaign and showed no further improvement with resolutions better than 5m. Tonttila et al.

(2021) show very small difference between 5 and 10 meter model vertical resolutions with UCLALES-SALSA model, although for a different case.

In the preliminary testing phase, we also investigated the model noise by performing two almost identical simulations that differed only by the random perturbations in the temperature field used to initialize the turbulence. The results of this experiment are shown on Figure 5. Expectedly, the noise is largest in the precipitation flux, while other quantities do not show large discrepancies, except for the cloud drop size at the very end of the simulation. Any simulation results differing from each other in similar level to what is shown on Figure 4 have been considered identical in the manuscript.

[Figure]

*Figure 5. Timeseries of two simulations that differ only by the random perturbations used to initialize the turbulence.*

The results of the above-mentioned sensitivity tests have been included to the Supplementary Information of the manuscript.

While the paper investigates one pathway or the other, it is unclear what would happen if both pathways were used (presumably the most realistic setup?). Please discuss.

We expect the impact of the pathways that are discussed to be very close to additive. For instance, some of our preliminary simulations not shown in the paper demonstrated that the effects of the additional CCN from the Fuentes et al. (2010) parameterization and the reduced hygroscopicity due to the organic fraction mostly cancel each other out. On the other hand, the effects of SOA formation would be smaller when including sea spray emission as in higher aerosol load it would have larger number of particles to condense on and thus individual particles would grow less.

We have added a paragraph discussing this aspect to the Discussion section of the manuscript.

The term "cloud lifetime" is used throughout the paper. Do the authors refer to a single cloud cell? If so, please define so upon first use. Else (if the cloud fraction of the entire cloud deck is meant) please rephrase "lifetime" to "fraction" and perhaps also show cloud fraction.

Following the suggestion of Reviewer #1, we now use the term "transitioning timescale between closed and open cells" instead of "cloud lifetime". Cloud fraction has been added to the figures.

Minor concerns

l. 10 "in larger scale" should be rephrased.

The sentence was rephrased.

l. 63 "as shown by (…)" should have not parenthesis.

The reference style was corrected.

ll. 76-83 Perhaps also provide actual kappa values in addition to qualitative adjectives (i.e., "high" and "low").

Kappa values have been added.

ll. 118ff. It is unclear which category (of the aforementioned overview) isoprene and monoterpenes fall into. Please clarify/expand.

Isoprene and monoterpenes fall into the terpenoid category. Clarifying sentence has been added.

l. 130 "DOC" is defined for a second time and differently from the first time (l. 72).

Definition has been corrected.

l. 148 "wind" should be "windspeed"?

The text has been changed.

l. 175 Please provide a citation.

Our approach deviates slightly from the classical implementations of Volatility Basis Set (VBS) (e.g. Farina et al. (2010)) that assume instant equilibrium between the gas and aerosol phases. In particular, we first use the VBS framework to compute the equilibrium vapour pressure of each semivolatile species for every aerosol, cloud and precipitation size bin according to their organic content. These values are then used for computing diffusion-limited condensation and evaporation using the same scheme as is used the model for water vapor, described in detail by Tonttila et al. (2017).

The model description has been amended.

ll. 175-176 It is unclear how aging could be relevant here. Please briefly explain.

In Volatility Basis Set models aging usually refers to oxidation of the semivolatile species further reducing their volatility. Including this effect would in time increase the SOA amount. However, our simulation is relatively short. Also, according to Farina et al., (2010) and Lane et al., (2008), it is unclear whether it is correct to include this effect for the biogenic VOCs as it is likely to be at least partly included in the applied SOA yields.

For clarity, the term "aging" has been replaced with "further oxidation of the semivolatile species".

l. 195 Please explain "dry-size based cloud bins" as this term appears contradictory.

The aerosol model in UCLALES-SALSA is a sectional model where the aerosol particle bins are based on dry diameter of the particles. This description is extended to also cover the cloud droplets so that particles that activate as cloud droplets are moved to a parallel bin structure identical to the one for aerosols. Thus we can have both droplets and aerosol particles from the same sized bins. For every bin (aerosol and cloud) the number concentration, dry amount of all chemical species and water amount are tracked. Only after the cloud droplets are assumed to form precipitation, they are moved to separate rain bins where the bin limits are defined by the wet size. This is explained in detail by Tonttila et al. (2017).

Clarification has been added to the manuscript.

l. 247 It is not clear where the authors see "open cells". Perhaps rephrase/clarify.

The open cells are indeed too large to be visible in the 10 km domain. The open cell structure is visible in Fig. 3 that shows the cloud pattern in the preliminary 20x20 km domain simulation mentioned above.

The figure has been included to the Supplemental Information.

l. 290 Please elaborate on how reduced fine mode aerosol (presumably too small to serve as CCN) reduce "cloud lifetime".

The size range that is reduced for the warmer temperatures reaches up to ~0.1 micron dry diameter. As seen from the Figure 11, panel b in the manuscript, particles smaller than that make up a noticeable fraction (~20%) of the activated cloud droplets.

A paragraph discussing the impact of aerosol particles of different sizes was added to the Discussion section.

l. 321 Why was DOC set to zero?

Our idea was to apply all the effects one by one in order to compare their magnitudes. The explanation was added to the paper.

l. 365 Is this a realistic setup (no sea spray, just secondary formation)?

While the reviewer is correct to point out that such conditions are not the most common, the gaseous VOC and particulate sea spray emissions do have somewhat different drivers. Especially, sea spray emission has much stronger dependence on windspeed, and thus the emission can be minuscule in quieter conditions. However, the simulations set up to show one effect at time give us the opportunity to directly compare the magnitude of the different effects. A paragraph discussing the additivity of the effects has been included to the Discussion section.

ll. 409-410 As currently written, it is unclear which one is zero and which one isn't.

The sentence has been clarified.

Fig. 1 Please briefly explain how dry size was obtain from size at RH~80%.

The emission fluxes in Figure 1 are shown for pure sea salt. For sea salt the dry diameter is assumed to be half of the one at 80% RH. The error of this relation was reported to be below 5% by Sofiev et al. (2011). This has been clarified in the paper.

**References**

Ackerman, A. S., van Zanten, M. C., Stevens, B., Savic-Jovcic, V., Bretherton, C. S., Chlond, A., Golaz, J. C., Jiang, H., Khairoutdinov, M., Krueger, S. K., Lewellen, D. C., Lock, A., Moeng, C. H., Nakamura, K., Petters, M. D., Snider, J. R., Weinbrecht, S. and Zulauf, M.: Large-eddy simulations of a drizzling, stratocumulus-topped marine boundary layer, Mon. Weather Rev., 137(3), 1083–1110, doi:10.1175/2008MWR2582.1, 2009.

Calderón, S. M., Tonttila, J., Buchholz, A., Joutsensaari, J., Komppula, M., Leskinen, A., Hao, L., Moisseev, D., Pullinen, I., Tiitta, P., Xu, J., Virtanen, A., Kokkola, H. and Romakkaniemi, S.: Aerosol-stratocumulus interactions : Towards a better process understanding using closures between observations and large eddy simulations, Atmos. Chem. Phys. Discuss. [preprint], in review, doi:https://doi.org/10.5194/acp-2022-273, 2022.

Clark, T. L.: Numerical Modeling of the Dynamics and Microphysics of Warm Cumulus Convection, J. Atmos. Sci., 30, 857–878, 1973.

Dror, T., Michel Flores, J., Altaratz, O., Dagan, G., Levin, Z., Vardi, A. and Koren, I.: Sensitivity of warm clouds to large particles in measured marine aerosol size distributions-a theoretical study, Atmos. Chem. Phys., 20(23), 15297–15306, doi:10.5194/acp-20-15297-2020, 2020.

Farina, S. C., Adams, P. J. and Pandis, S. N.: Modeling global secondary organic aerosol formation and processing with the volatility basis set: Implications for anthropogenic secondary organic aerosol, J. Geophys. Res. Atmos., 115(9), 1–17, doi:10.1029/2009JD013046, 2010.

Fuentes, E., Coe, H., Green, D., De Leeuw, G. and McFiggans, G.: On the impacts of phytoplankton-derived organic matter on the properties of the primary marine aerosol - Part 1: Source fluxes, Atmos. Chem. Phys., 10(19), 9295–9317, doi:10.5194/acp-10-9295-2010, 2010.

Gong, S. L.: A parameterization of sea-salt aerosol source function for sub- and super-micron particles, Global Biogeochem. Cycles, 17(4), 1–7, doi:10.1029/2003gb002079, 2003.

Kazil, J., Yamaguchi, T. and Feingold, G.: Mesoscale organization, entrainment, and the properties of a closed-cell stratocumulus cloud, J. Adv. Model. Earth Syst., 9, 2214–2229, doi:10.1002/2017MS001072, 2017.

Kim, M. J., Novak, G. A., Zoerb, M. C., Yang, M., Blomquist, B. W., Huebert, B. J., Cappa, C. D. and Bertram, T. H.: Air-Sea exchange of biogenic volatile organic compounds and the impact on aerosol particle size distributions, Geophys. Res. Lett., 44(8), 3887–3896, doi:10.1002/2017GL072975, 2017.

Lane, T. E., Donahue, N. M. and Pandis, S. N.: Simulating secondary organic aerosol formation using the volatility basis-set approach in a chemical transport model, Atmos. Environ., 42(32), 7439–7451, doi:10.1016/j.atmosenv.2008.06.026, 2008.

Seifert, A. and Beheng, K. D.: A double-moment parameterization for simulating autoconversion, accretion and selfcollection, Atmos. Res., 59–60, 265–281, doi:10.1016/S0169-8095(01)00126-0, 2001.

Shima, S., Kusano, K., Kawano, A., Sugiyama, T. and Kawahara,  and S.: The super-droplet method for the numerical simulation of clouds and precipitation: A particle-based and probabilistic microphysics model coupled with a non-hydrostatic model, Q. J. R. Meteorol. Soc., 135, 1307–1320, doi:10.1002/qj.441, 2009.

Sofiev, M., Soares, J., Prank, M., De Leeuw, G. and Kukkonen, J.: A regional-to-global model of emission and transport of sea salt particles in the atmosphere, J. Geophys. Res. Atmos., 116(21), doi:10.1029/2010JD014713, 2011.

Stevens, B., Moeng, C. H., Ackerman, A. S., Bretherton, C. S., Chlond, A., de Roode, S., Edwards, J., Golaz, J. C., Jiang, H., Khairoutdinov, M., Kirkpatrick, M. P., Lewellen, D. C., Lock, A., Müller, F., Stevens, D. E., Whelan, E. and Zhu, P.: Evaluation of large-eddy simulations via observations of nocturnal marine stratocumulus, Mon. Weather Rev., 133(6), 1443–1462, doi:10.1175/MWR2930.1, 2005.

Tonttila, J., Maalick, Z., Raatikainen, T., Kokkola, H., Kühn, T. and Romakkaniemi, S.: UCLALES-SALSA v1.0: A large-eddy model with interactive sectional microphysics for aerosol, clouds and precipitation, Geosci. Model Dev., 10(1), 169–188, doi:10.5194/gmd-10-169-2017, 2017.

Tonttila, J., Afzalifar, A., Kokkola, H., Raatikainen, T., Korhonen, H. and Romakkaniemi, S.: Precipitation enhancement in stratocumulus clouds through airborne seeding: Sensitivity analysis by UCLALES-SALSA, Atmos. Chem. Phys., 21(2), 1035–1048, doi:10.5194/acp-21-1035-2021, 2021.